# PSP: Prompt-Guided Self-Training Sampling Policy for Active Prompt Learning

**Sen Tao**[1,2,3,4], **Kaiduo Feng**[1,3,4], **Jiawei Liu**[2]*, **Peng Zeng**[1,3]*, **Yongchao Xu**[2],
Yufei Zheng[2], Zheng-Jun Zha[2]

[1]State Key Laboratory of Robotics, Shenyang Institute of Automation, CAS, Shenyang, China
[2]University of Science and Technology of China, Hefei, China
[3]Key Laboratory of Networked Control Systems, Shenyang Institute of Automation, China
[4]University of Chinese Academy of Sciences, Beijing, China

`taosen23@mails.ucas.ac.cn, {jwliu6, zhazj}@ustc.edu.cn, zp@sia.cn`

## Abstract

Active Prompt Learning (APL) using vision-language models (*e.g.*, CLIP) has attracted considerable attention for mitigating the dependence on fully labeled dataset in downstream task adaptation. However, existing methods fail to explicitly leverage prompt to guide sample selection, resulting in the selected samples being ineffective in facilitating the prompt template's downstream task adaptation, while also overlooking valuable complementary information in the unselected samples. To fill this gap, we propose a novel Prompt-Guided Self-Training Sampling Policy (PSP) for APL, which integrates Soft Actor-Critic with a customized real-pseudo hybrid reward and vectorized critics to incorporate prompts in guiding sample selection toward those that facilitate the optimization of prompt template, by jointly considering both selected and unselected samples. Specifically, PSP comprises two prominent components: Vectorized Soft Actor-Critic Sampling Policy (VSSP) and Uncertainty Augmented Self-Training (UST) mechanism. VSSP customizes a real-pseudo hybrid reward based on learned prompts and image features, which is fed into vectorized critics to estimate Q-value for each sample and compute gradients that optimize the actor, allowing it to refine its sampling policy in an End-to-End manner to identify the most informative samples for prompt learning. Moreover, UST leverages the CLIP from the previous round to generate reliable pseudo-labeled data based on uncertainty and confidence of average predictions, thereby deepening the understanding of the overall data. Extensive experiments conducted on diverse real-world datasets validate the effectiveness of our PSP.

## 1 Introduction

Recent research in pre-trained Vision-Language Models (VLMs) has demonstrated impressive performance across various tasks, largely through prompt learning that fine-tunes a small set of parameters within a learnable prompt on fully labeled dataset. For instance, Contrastive Language-Image Pre-training (CLIP) (Radford et al., 2021) is a representative model that consists of image and text encoders using a contrastive loss function, trained on 0.4 billion text-image pairs, and is renowned for its robust transferability. Building on CLIP (Radford et al., 2021), Zhou *et al.* proposed CoOp (Zhou et al., 2022b), a notable approach that freezes both the image and text encoders, enabling learnable context vectors to serve as templates. However, the resource consumption required for annotation remains substantial.

In response, some researchers have turned to active learning, which selects the most informative samples within a limited annotation budget to maximize performance (Xie et al., 2023). The core challenge of active learning lies in formulating an effective criterion for sample selection. Conventional active learning methods are divided into three categories based on the sampling

---
*Corresponding authors.

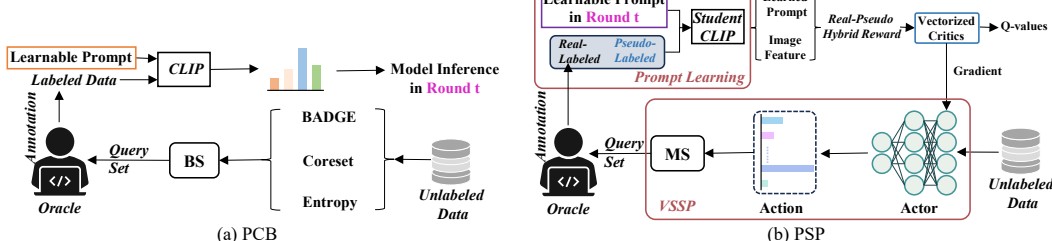

Figure 1: **Illustration of (a) PCB (Bang et al., 2024), (b) Our PSP.** PSP introduces the Vectorized Soft Actor-Critic Sampling Policy (VSSP) to replace the conventional sampling algorithm in PCB while filtering reliable pseudo-labeled data through UST.

algorithm: (1) Uncertainty-based sampling (Lewis & Catlett, 1994; Gal et al., 2017; Kirsch et al., 2019; Holub et al., 2008) selects the most uncertain samples, such as Entropy (Holub et al., 2008). (2) Diversity-based sampling prioritizes ensuring that the queried samples represent the entire data distribution, as in Clustering (Hu et al., 2021) and Coreset (Sener & Savarese, 2018). (3) Hybrid sampling aims to query informative samples by jointly considering uncertainty and diversity, such as BADGE (Ash et al., 2020), ALFA-Mix (Parvaneh et al., 2022) and GCNAL (Caramalau et al., 2021). (4) RL-based sampling formulates sample selection as a policy optimization problem, leveraging Reinforcement Learning (RL) to iteratively refine this policy and select samples that maximize the cumulative reward, such as DRAL (Liu et al., 2019) and AOL (Woodward & Finn, 2017).

These approaches generally rely on smaller foundational models, such as ResNet (He et al., 2016), which lack the commonsense knowledge and specialized domain expertise that larger models possess. Therefore, Bang *et al.* (Bang et al., 2024) introduced the pre-trained CLIP into active learning and proposed Active Prompt Learning (APL), along with the Pseudo-Class Balance (PCB) framework. As shown in Figure 1(a), PCB mechanically applies conventional sampling methods (*i.e.*, Entropy, Coreset, and BADGE) to select candidates, which are then sent to a Balance Sampler (BS) to create query set by preferentially selecting candidates whose pseudo-labels correspond to the most under-represented classes in labeled data. However, three PCB variants isolate sample selection and prompt learning in the APL task, lacking an explicit connection between the two, which is conceptually inappropriate. Additionally, three PCB variants also overlook complementary information in unselected samples, limiting further improvements in model performance. Therefore, in the context of APL, we urgently need a method that bridges these two stages by explicitly leveraging prompt to guide sample selection and fully exploiting the complementary information in unselected samples.

To address these issues, we propose a novel Prompt-Guided Self-Training Sampling Policy (PSP) for APL, which combines Soft Actor-Critic (SAC) (Haarnoja et al., 2018) with a tailored real-pseudo hybrid reward and vectorized critics to integrate prompts in directing sample selection toward those that advance the optimization of prompt template, by jointly considering both selected and unselected samples, as shown in Figure 1(b). Specifically, PSP establishes a self-training teacher-student framework composed of two key components: Vectorized Soft Actor-Critic Sampling Policy (VSSP) and Uncertainty Augmented Self-Training (UST) mechanism. VSSP first designs an actor to map the gradient embeddings of samples from unlabeled data pool into action, where each element represents the probability of selecting a given unlabeled sample. Next, VSSP utilizes Multinomial Sampling (MS) to construct the query set in round $t$, acquires real-labeled data through Oracle annotation, and combines it with pseudo-labeled data from UST for the student CLIP's prompt learning. After prompt learning, the learned prompts and image features are integrated into the computation of the real-pseudo hybrid reward, which is then passed to vectorized critics to estimate the Q-value for each sample and derive the actor's gradients, enabling it to refine its sampling policy in an End-to-End fashion and effectively identify the most informative samples for further enhancement. To harness the complementary information in unselected samples, UST employs the teacher CLIP model from round $t-1$ to generate reliable pseudo-labels by evaluating the uncertainty and confidence of the average predictions. Extensive experiments on multiple datasets prove the effectiveness of our PSP.

The main contribution of this work can be summarized as follows: (*i*) We propose a novel Prompt-Guided Self-Training Sampling Policy for active prompt learning, combining SAC with a customized real-pseudo hybrid reward and vectorized critics to guide sample selection towards those that promote the optimization of prompt template. (*ii*) We construct the Vectorized Soft Actor-Critic Sampling

Policy, which tailors a real-pseudo hybrid reward based on learned prompts and image features, feeding it into vectorized critics to compute gradients of the actor, allowing it to refine its sampling policy and identify the most informative samples for prompt learning. (*iii*) We develop an Uncertainty Augmented Self-Training mechanism, which generates reliable pseudo-labeled data based on the uncertainty and confidence of the average predictions to reveal data structures not reflected in the real-labeled data.

## 2 RELATED WORK

**Active Learning** identifies criteria for selecting the most informative samples under a limited labeling budget. Based on the criteria, active learning methods can be categorized into three main approaches: Uncertainty-based sampling (Gal et al., 2017; Wang et al., 2019), Diversity-based sampling (Hacohen et al., 2022; Shui et al., 2020), Hybrid sampling (Ash et al., 2020; Parvaneh et al., 2022; Caramalau et al., 2021), and RL-based sampling (Ash et al., 2020; Kirsch et al., 2019). For Uncertainty-based sampling, Entropy (Holub et al., 2008) selects the samples with the highest entropy for annotation on object recognition. For Diversity-based sampling, Coreset (Sener & Savarese, 2018) provides an approximate upper bound on the loss for feature space coverage-based active learning algorithms. For Hybrid Sampling, ALFA-Mix (Parvaneh et al., 2022) utilizes unlabeled data to support active learning by interpolating between the representations of labeled and unlabeled instances and identifying features the model fails to recognize through inconsistencies in predicted labels. For RL-based sampling, AOL (Woodward & Finn, 2017) combines meta-learning and reinforcement learning for one-shot classification tasks. DRAL (Liu et al., 2019) designs an agent in acquiring pairwise annotated data. Notably, PAL (Fang et al., 2017) builds a deep Q-network as an adaptive policy for sample selection. Therefore, we believe that RL-based methods have the potential to incorporate prompts for guiding sample selection. However, AOL (Woodward & Finn, 2017) and PAL (Fang et al., 2017) model the decision of whether to annotate a streaming unlabeled sample as a binary classification problem, while MedSelect (Vrabac et al., 2022) and DARL (Liu et al., 2019) rely on pairwise data, making them unsuitable for direct application in Active Prompt Learning (APL). Therefore, we introduce Soft Actor-Critic (SAC) (Haarnoja et al., 2018), a representative reinforcement learning algorithm known for its robustness to hyperparameters and strong performance in continuous action spaces. By designing a customized real-pseudo hybrid reward and vectorized critics, SAC can be seamlessly integrated into APL.

**Vision-Language Models** have recently demonstrated remarkable advancements in downstream tasks, leveraging their robust transfer learning capabilities. A prominent example is CLIP (Radford et al., 2021), which has been extensively adopted across a wide range of downstream applications (Yu et al., 2023; Zhou et al., 2023; Ning et al., 2023; Liang et al., 2023; Jia et al., 2022). Inspired by prompt optimization in natural language processing (Jiang et al., 2020; Khattak et al., 2023), CoOp (Zhou et al., 2022b) is a representative approach that transforms context words into learnable context vectors via a text encoder. Building on this, CoCoOp (Zhou et al., 2022a) further refines the learnable prompt by adapting it to individual image instances.

**Active Prompt Learning** (APL) resolves the dilemma between the need for additional labeled data to enhance prompt learning and the high cost of data annotation. It annotates valuable samples for prompt learning within a fixed budget, improving performance on downstream tasks. Furthermore, Bang *et al.* (Bang et al., 2024) proposed the Pseudo-Class Balance (PCB) framework for APL, which employs selection algorithms such as Entropy (Holub et al., 2008), Coreset (Sener & Savarese, 2018), and BADGE (Ash et al., 2020) to identify candidates. These candidates are then passed by a balance sampler to select candidates whose pseudo-labels correspond to the most underrepresented classes in the labeled data. Notably, Entropy is prone to noisy data and outliers, while Coreset prioritizes diversity but also includes less informative samples, and BADGE relies on fixed rules to balance diversity and uncertainty, limiting its adaptability across tasks. More importantly, these works treat sample selection and prompt learning as two decoupled, discrete stages, with a lack of explicit connection between them, which makes the APL task fragmented. In contrast, our method bridges these two stages by refining the sample policy through a customized reward derived from the prompt learning process, thereby explicitly leveraging the prompt to guide sample selection.

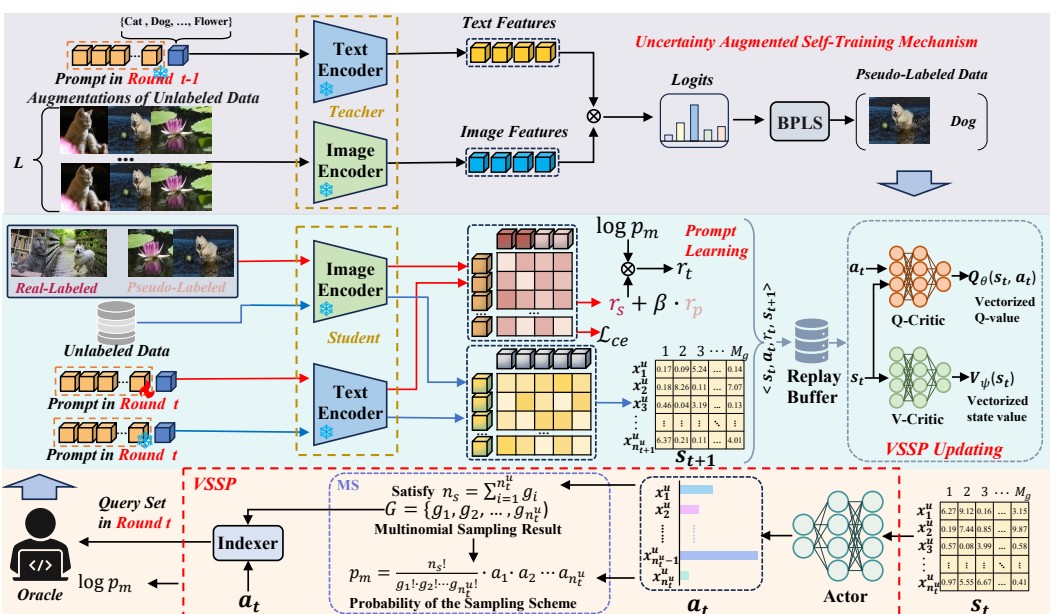

Figure 2: **The overall structure of our PSP.** The CLIP collaborative learning framework for PSP consists of two core components: the Vectorized Soft Actor-Critic Sampling Policy (VSSP) and the Uncertainty Augmented Self-Training (UST) mechanism.

## 3 METHODOLOGY

### 3.1 OVERVIEW

PSP comprises two crucial components: the Vectorized Soft Actor-Critic Sampling Policy (VSSP) and the Uncertainty Augmented Self-Training (UST) mechanism, as described in Figure 2. VSSP employs the Actor network to map the gradient embeddings of $n_t^u$ unlabeled samples into action $\boldsymbol{a}_t = \{a_1, a_2, \ldots, a_{n_t^u}\}$, where each element indicates the probability of unlabeled sample being selected. Next, VSSP utilizes Multinomial Sampling (MS) to derive sampling results $G = \{g_1, g_2, \ldots, g_{n_t^u}\}$, which are combined with $\boldsymbol{a}_t$ and then fed into the Indexer to index the query set in round $t$. MS additionally outputs the MS indicator $\log p_m$ to reflect the quality of the sampling scheme. Ultimately, the Oracle is utilized to annotate the query set, thereby yielding the real-labeled data. Simultaneously, UST leverages the teacher CLIP $\mathcal{F}^t(\mathcal{F}_V^t$ and $\mathcal{F}_T^t)$ to obtain the pseudo-labels for the remaining unlabeled data. To mitigate the disruptive effects of noise in pseudo-labels, UST introduces the Balanced Pseudo-Labeled Selective (BPLS) module, which filters out balanced and reliable pseudo-labeled data by jointly evaluating the uncertainty and confidence of the average predictions across $L$ augmentations. Real-labeled data is combined with pseudo-labeled data for the student CLIP $\mathcal{F}^s(\mathcal{F}_V^s$ and $\mathcal{F}_T^s)$ in prompt learning, optimized with cross-entropy loss $\mathcal{L}_{ce}$. In prompt learning, the text prompt $\boldsymbol{p}_c$ for class $c$ is constructed by appending the class token $[\text{cls}_c]$ at the end of a sequence, as illustrated below:

$$\boldsymbol{p}_c = [\boldsymbol{c}]_1 [\boldsymbol{c}]_2 \ldots [\boldsymbol{c}]_M [\text{cls}_c] \tag{1}$$

where $[\boldsymbol{c}]_m$ indicates the learnable context vector of the prompt $\boldsymbol{p}_c$, with dimensions matching those of the word embeddings, while $M$ represents the prompt size. We design experiments regarding learnable prompt analysis in Appendix A.2. Following PCB, we incorporate class-specific descriptions generated by GPT-3 (Brown et al., 2020) for augmentation to further enhance prompt learning, as detailed in Appendix A.4. After prompt learning, the real-pseudo hybrid reward $r_t$ is computed by the product of $\log p_m$ and the sum of a real-labeled reward $r_s$ and a pseudo-labeled reward $r_p$. Since the unlabeled data pool excludes the query set, the next state $\boldsymbol{s}_{t+1}$ is calculated by feeding $n_{t+1}^u$ unlabeled samples into the student CLIP after prompt learning. At the end of each round, experience tuple $\langle \boldsymbol{s}_t, \boldsymbol{a}_t, r_t, \boldsymbol{s}_{t+1} \rangle$ is stored in replay buffer, which is subsequently used to update the networks in VSSP.

## 3.2 VECTORIZED SOFT ACTOR-CRITIC SAMPLING POLICY

VSSP introduces prompt to guide sampling policy in selecting the most informative samples for prompt learning, thereby bridging sample selection and prompt learning—two stages that previous works have treated in isolation. Next, we provide a detailed elaboration of our VSSP.

We define APL by the tuple $< S, A, p_s, r >$, where the state space $S$ and the action space $A$ are continuous, the state transition probability $p_t : S \times S \times A \to [0, +\infty]$ denotes the probability density of the next state $\boldsymbol{s}_{t+1}$ given the current state $\boldsymbol{s}_t$ and action $\boldsymbol{a}_t$, and the reward $r : S \times A \to \mathbb{R}$. We also adopt a vectorized state value function $\boldsymbol{V}_\psi(\boldsymbol{s}_t)$, a vectorized Q-function $\boldsymbol{Q}_\theta(\boldsymbol{s}_t, \boldsymbol{a}_t)$, and a sampling policy $\pi_\phi(\boldsymbol{a}_t | \boldsymbol{s}_t)$, with network parameters $\psi$, $\theta$, and $\phi$, inspired by SAC (Haarnoja et al., 2018). The Q-Critic and V-Critic networks are modeled as fully connected networks, while the sampling policy is built as a Gaussian distribution, with its mean and covariance predicted by the Actor network.

**State.** Given the unlabeled data pool $\mathcal{D}_u$, VSSP defines the state $\boldsymbol{s}_t \in \mathbb{R}^{n_t^u \times M_g}$ as a matrix representing the gradient embeddings of the $n_t^u$ unlabeled samples. Here, $M_g = K \times D_V^t$ denotes the dimension of the gradient embeddings, $K$ indicates the total number of classes and $D_V^t$ is the dimension of the teacher image features $\boldsymbol{f}_V^{t,i} = \mathcal{F}_V^t(\boldsymbol{x}_i^u)$. Gradient embedding incorporates prompt information to enrich the state representation, offering richer gradient insights during sampling policy updating compared to a single image feature. The analysis of state modeling is executed in Section 4.3. Formally, the $i$-th value of $\boldsymbol{s}_t$ is expressed below:

$$\boldsymbol{s}_t^i = \begin{cases} \boldsymbol{f}_V^{t,i} \cdot [1 - \cos(\mathcal{F}_T^t(\boldsymbol{p}_c), \boldsymbol{f}_V^{t,i})], & \text{if } c = \hat{y}_i \\ -\boldsymbol{f}_V^{t,i} \cdot \cos(\mathcal{F}_T^t(\boldsymbol{p}_c), \boldsymbol{f}_V^{t,i}), & \text{if } c \neq \hat{y}_i \end{cases} \tag{2}$$

Here, $\cos(\mathcal{F}_T^t(\boldsymbol{p}_c), \boldsymbol{f}_V^{t,i})$ denotes the score that an unlabeled sample $\boldsymbol{x}_i^u$ belongs to class $c$, for $c = 1, 2 \ldots, K$. $\hat{y}_i$ represents predicted category of unlabeled sample $\boldsymbol{x}_i^u$.

**Action.** We define the action as a vector $\boldsymbol{a}_t \in \mathbb{R}^{n_t^u}$, where each element represents the probability of the unlabeled sample selected by the actor. The sampling policy generates the action $\boldsymbol{a}_t \in \mathbb{R}^{n_t^u}$ based on the current state $\boldsymbol{s}_t$. After obtaining the action vector, VSSP adopts the Multinomial Sampling (MS) to obtain the query set, which introduces randomness and helps to distribute the selected samples more evenly. The sampling results $G = \{g_1, g_2, \ldots, g_{n_t^u}\}$ in MS follows a Multinomial Distribution, where $g_i$ denotes the number of times $\boldsymbol{x}_i^u$ is selected, and satisfying $n_s = \sum_{i=1}^{n_t^u} g_i$. Therefore, VSSP defines the log probability of the sampling scheme as MS indicator to evaluate the quality of the sampling scheme.

$$\log(p_m(g)) = \log\left(\frac{n_s!}{g_1! \cdot g_2! \cdot \cdots \cdot g_{n_t^u}!}\right) + \sum_{i=1}^{n_t^u} g_i \log a_i \tag{3}$$

A larger value of $\log(p_m(g))$ indicates that the current scheme aligns well with the distribution of $\boldsymbol{a}_t$, and vice versa. The sampling results and action are fed into the Indexer to retrieve the selected samples, replacing any duplicates with the sample that has the higher probability. Hence, VSSP obtains the query set $\{\boldsymbol{x}_i^s\}_{i=1}^{n_s}$, which is then presented to the Oracle, resulting in the labeled set $\{\boldsymbol{x}_i^s, y_i^s\}_{i=1}^{n_s}$. The real-labeled data are indicated as $\mathcal{D}_l = \mathcal{D}_l \cup \{\boldsymbol{x}_i^s, y_i^s\}_{i=1}^{n_s}$. Meanwhile, the unlabeled data pool will exclude the labeled set, *i.e.*, $\mathcal{D}_u \setminus \{\boldsymbol{x}_i^s\}_{i=1}^{n_s}$.

**Shape-Variable State Transition.** The real-labeled data $\mathcal{D}_l$ is incorporated with the pseudo-labeled data $\mathcal{D}_p = \{\boldsymbol{x}_i^p, \hat{y}_i^p\}_{i=1}^{n_p}$ for the student's prompt learning with a cross-entropy loss. After prompt learning, the next state $\boldsymbol{s}_{t+1}$ is obtained by feeding the remaining $n_{t+1}^u$ unlabeled samples into the student CLIP, where $n_{t+1}^u = n_t^u - n_s$. Consequently, the state transition exhibits a variable shape, leading to alignment issues in Equation 8 when optimizing the Q-Critic network, distinguishing it from SAC (Haarnoja et al., 2018).

**Reward.** To explicitly utilize prompt for guiding sample selection, we first propose the real-pseudo hybrid reward, inspired by (Wang et al., 2020), as outlined below.

$$r(\boldsymbol{s}_t, \boldsymbol{a}_t) = \log(p_m(g)) * (\overline{\boldsymbol{r}}_s + \beta \overline{\boldsymbol{r}}_p) \tag{4}$$

$$r_k^i = \max_{c=1}^K \cos(\mathcal{F}_T^s(\boldsymbol{p}_c), \mathcal{F}_V^s(\boldsymbol{x}_i^k)) - \cos(\mathcal{F}_T^s(\boldsymbol{p}_{y_i^k}), \mathcal{F}_V^s(\boldsymbol{x}_i^k)) \tag{5}$$

where $r_k^i$ denotes the reward for a single sample, reflecting the quality of individual prediction, $k \in \{s, p\}$ indicates the data type, where $(x_i^s, y_i^s)$ denotes the real-labeled sample with label $y_i^s$, and

$(x_i^p, y_i^p)$ denotes the pseudo-labeled sample with pseudo label $y_i^p = \hat{y}_i^u$, where $\hat{y}_i^u$ is the prediction of the teacher CLIP model, and coefficient $\beta$ represents the contribution of pseudo-labeled rewards. Notably, $\overline{r}_s$ indicates the mean of vector $r_s$. Considering that MS indicator $\log(p_m(g))$ is negative while $r_s$ and $r_p$ are positive, maximizing the real-pseudo hybrid reward results in a reduction of $r_s + \beta r_p$, which improves the model's classification ability. $\log(p_m(g))$ is closely related to the construction of the query set for labeling and conveys information for the subsequent training of the actor and critic, reflecting the quality of the sampling scheme.

**Training.** As shown in Algorithm 1, each round stores an experience in the replay buffer. Given the limited number of experiences, VSSP trains the actor and critics by sampling a single experience per gradient step once the number of stored experiences exceeds the buffer threshold $\tau_b$, with the analysis of $\tau_b$ are provided in Figure 5 from Appendix A.2. In APL, the selection of each sample is made independently. However, using a scalar state value and Q-value assigns a shared global information value to all samples, preventing the differentiation of individual sample contributions for sampling policy updating. To address this, VSSP employs a vectorized V-Critic and Q-Critic to estimate the state value and Q-value for each sample, enabling finer-grained control over each sample's role in optimizing the sampling policy. The V-Critic network updates its parameters by minimizing the following squared residual error.

$$J_V(\psi) = \mathbb{E}_{s_t \sim \mathcal{B}} \left[ \frac{1}{2} \| V_\psi(s_t) - U_t^V \|_2^2 \right] \tag{6}$$

where $\mathcal{B}$ denotes a replay buffer that stores history experiences, $U_t^V = \mathbb{E}_{a_t \sim \pi_\phi} [Q_\theta(s_t, a_t) - \log \pi_\phi(a_t|s_t)]$ indicates the target value for training the V-Critic network. $Q_\theta(s_t, a_t) \in \mathbb{R}^{n_t^u}$ represents the Q-value predicted by the Q-Critic network for the current state $s_t \in \mathbb{R}^{n_t^u \times Mg}$ and action $a_t$, and $V_\psi(s_t) \in \mathbb{R}^{n_t^u}$ indicates the state value predicted by the V-Critic network. The unbiased estimates of the gradient of Equation 6 are computed as below:

$$\hat{\nabla}_\psi J_V(\psi) = (V_\psi(s_t) - Q_\theta(s_t, a_t) + \log \pi_\phi(a_t|s_t))^\top \nabla_\psi V_\psi(s_t) \tag{7}$$

Here, the action $a_t$ is drawn according to the current policy $\pi_\phi(\cdot|s_t)$, instead of the replay buffer $\mathcal{B}$. $\log \pi_\phi(a_t|s_t)$ is broadcasted to match the shape of $V_\psi(s_t)$. Similarly, the Q-Critic network is optimized to minimize the modified soft Bellman residual:

$$J_Q(\theta) = \mathbb{E}_{(s_t, a_t) \sim \mathcal{B}} \left[ \frac{1}{2} \| Q'_\theta(s_t, a_t) - \hat{Q}'(s_t, a_t)) \|_2^2 \right] \tag{8}$$

where $Q'_\theta, \hat{Q}' = W(Q_\theta, \hat{Q})$ indicates the aligned target Q-value and Q-value after alignment through Soft Dynamic Time Warping (Soft-DTW) (Cuturi & Blondel, 2017), as detailed in Appendix A.5. We adopt Soft-DTW for alignment because it preserves the relative order and structural relationships between elements, which is crucial given the strong correlation between target Q-value and Q-value. In addition, the differentiability of Soft-DTW enables gradients to propagate through the alignment process, ensuring that it does not interfere with the update procedures of either the Actor or the Critic.

The target Q-value $\hat{Q} \in \mathbb{R}^{n_t^u}$ is defined below:

$$\hat{Q}(s_t, a_t) = r(s_t, a_t) + \gamma \mathbb{E}_{s_{t+1} \sim p} \left[ V_{\overline{\psi}}(s_{t+1}) \right] \tag{9}$$

where $\gamma$ denotes the discount factor, $V_{\overline{\psi}}(s_{t+1}) \in \mathbb{R}^{n_{t+1}^u}$ represents the state value predicted by the target V-Critic network with parameters $\overline{\psi}$. The scalar reward $r(s_t, a_t) \in \mathbb{R}$ is broadcasted to match the shape of $V_{\overline{\psi}}(s_{t+1})$.

The state is shape-variable, causing $Q_\theta$ and $\hat{Q}$ to have mismatched dimensions, making direct subtraction infeasible in the soft Bellman residual of SAC. Soft-DTW addresses this issue by optimizing a smooth, differentiable relaxation of the optimal matching cost, ensuring that $Q'_\theta$ is a differentiable function and preventing information loss result from cropping or padding. Consequently, the stochastic gradient of Equation 8 is illustrated below:

$$\hat{\nabla}_\theta J_Q(\theta) = \left( Q'_\theta(s_t, a_t) - \hat{Q}'(s_t, a_t) \right)^\top \nabla_\theta Q'_\theta(s_t, a_t) \tag{10}$$

The parameter update uses a target V-Critic network $V_{\overline{\psi}}$, whose update is a weighted average, with a hyperparameter $\tau$ controlling the degree of mixing between the V-Critic network parameters $\psi$ and the target V-Critic network parameters $\overline{\psi}$.

**Algorithm 1** VSSP

1: Initialize parameter vectors $\psi, \overline{\psi}, \theta, \phi$.
2: **for** round $t$ in range $(R)$ **do**
3: $\quad a_t \sim \pi_\phi(a_t|s_t)$
4: $\quad s_{t+1} \sim p_s(s_{t+1}|s_t, a_t)$
5: $\quad \mathcal{B} \leftarrow \mathcal{B} \cup \{(s_t, a_t, r_t, s_{t+1})\}$
6: $\quad$ **for** each gradient step **do**
7: $\qquad \psi \leftarrow \psi - \lambda_V \hat{\nabla}_\psi J_V(\psi)$
8: $\qquad \theta \leftarrow \theta - \lambda_Q \hat{\nabla}_\theta J_Q(\theta)$
9: $\qquad \phi \leftarrow \phi - \lambda_\pi \hat{\nabla}_\phi J_\pi(\phi)$
10: $\qquad \psi \leftarrow \tau\psi + (1-\tau)\overline{\psi}$
11: $\quad$ **end for**
12: **end for**

**Algorithm 2** UST

**Initial:** $\mathcal{D}_p = \emptyset$
1: **for** each unlabeled sample **do**
2: $\quad \{x_i^{u,l}\}_{l=1}^L \leftarrow x_i^u$
3: $\quad z_i^l = \mathcal{F}^t(x_i^{u,l})$
4: $\quad \hat{y}_i^u = \arg\max z_i^{avg} = \arg\max \frac{1}{L}\sum_{l=1}^L z_i^l$
5: $\quad g_i^c = confidence(z_i^{avg})$
6: $\quad g_i^u = std\{confidence(z_i^l)\}_{l=1}^L$
7: $\quad$ **if** $g_i^c \geq \tau_c$ and $g_i^u \leq \tau_u$ **then**
8: $\qquad$ where $\tau_c = \frac{1}{B}\sum_{i=1}^B g_i^c$, $\tau_u = \frac{1}{B}\sum_{i=1}^B g_i^u$, and $B$ denotes batch size.
9: $\qquad \mathcal{D}_p \leftarrow \mathcal{D}_p \cup (x^p = x_i^u, y^p = \hat{y}_i^u)$
10: $\quad$ **end if**
11: **end for**

To ensure that VSSP effectively explores while maintaining stable and efficient convergence during training, we incorporate the reparameterization trick, which introduces stochasticity by reparameterizing the policy through the Actor network as follows:

$$a_t' = f_\phi(\epsilon_t; s_t) = f_\phi^\mu(s_t) + \epsilon_t \odot f_\phi^\sigma(s_t) \tag{11}$$

where $\epsilon_t$ is an input noise vector sampled from a standard normal distribution, $f_\phi^\mu(s_t)$ and $f_\phi^\sigma(s_t)$ indicates the mean and covariance predicted by the Actor network, respectively. The goal of the Actor network is to learn the policy that maximizes the following objective (Ziebart, 2010):

$$J_\pi(\phi) = \mathbb{E}_{s_t \sim \mathcal{D}, \epsilon_t \sim \mathcal{N}} \left[ \log \pi_\phi \left( f_\phi(\epsilon_t; s_t) \mid s_t \right) - \frac{1}{n_t^u} \sum_{i=1}^{n_t^u} Q_\theta^i(s_t, f_\phi(\epsilon_t; s_t)) \right] \tag{12}$$

Here, $\pi_\phi$ is defined as a Gaussian distribution based on the Actor network outputs $f_\phi^\mu(s_t)$ and $f\phi^\sigma(s_t)$. The gradient of Equation 12 can be approximated as follows:

$$\hat{\nabla}_\phi J_\pi(\phi) = -\frac{1}{n_t^u}\sum_{i=1}^{n_t^u} \nabla_{a_t'} Q_\theta^i(s_t, a_t')^\top \nabla_\phi f_\phi(\epsilon_t; s_t) + \nabla_\phi \log \pi_\phi(a_t'|s_t) + \nabla_{a_t'} \log \pi_\phi(a_t'|s_t)^\top \nabla_\phi f_\phi(\epsilon_t; s_t) \tag{13}$$

### 3.3 UNCERTAINTY AUGMENTED SELF-TRAINING

The UST mechanism is designed to uncover the latent, task-specific knowledge embedded in unlabeled data that prior works have overlooked, thereby enriching the model's understanding of the overall data distribution. Notably, we compare UST with other semi-supervised (Chakraborty et al., 2024) and unsupervised (Huang et al., 2022) prompt learning methods to further assess its effectiveness, as detailed in Table 4 from Appendix A.2. As outlined in Algorithm 2, UST begins by employing the same data augmentation to sample $x_i^u$ from remaining unlabeled data $\mathcal{D}_u \backslash \{x_i^s\}_{i=1}^{n_s}$ to generate $L$ augmentations of each unlabeled sample $\{x_i^{u,l}\}_{l=1}^L$. After freezing the learnable prompt from round $t-1$ to construct the teacher CLIP model, UST feeds $L$ augmentations of the unlabeled data into the teacher CLIP model $\mathcal{F}^t$ to compute logits for each augmented versions of the unlabeled samples $\{z_i^l\}_{l=1}^L$. To achieve stable predictions from the teacher CLIP model, UST calculates the average of the logits $z_i^{avg}$ to obtain the average prediction $\hat{y}_i^u$, which serves as the pseudo-label for the unlabeled sample $x_i^u$.

Considering that noise in pseudo-labels can interfere optimization process, UST designs a Balanced Pseudo-Label Selective (BPLS) module to filter out samples with more reliable pseudo-labels by jointly evaluating prediction uncertainty and confidence. In BPLS, $g_i^c$ represents the confidence score computed from the average logits $z_i^{avg}$ across augmentations of unlabeled data, while $g_i^u$ quantifies uncertainty using the standard deviation of confidence scores from different augmented versions. BPLS selects samples where $g_i^c$ exceeds the prediction confidence threshold $\tau_c$ and $g_i^u$ is below the prediction uncertainty threshold $\tau_u$. Samples meeting these criteria are incorporated into the reliable

pseudo-labeled data $\mathcal{D}_p$. After filtering all samples, UST identifies the missing categories in the filtered pseudo-labeled and selects high-confidence samples corresponding to those categories, filling the gaps until the number of samples matches the minimum class count in $\mathcal{D}_p$. Finally, pseudo-labeled data are combined with real-labeled data for the student CLIP's prompt learning, enabling a more comprehensive understanding of the intrinsic structure and relationships within the data.

## 4 EXPERIMENTS

### 4.1 EXPERIMENTAL SETTING

**Datasets & Metrics.** We adopt seven commonly used datasets to evaluate our PSP. These datasets encompass diverse categories and are sufficient to demonstrate that PSP can address various real-world scenarios, including Stanford Cars (Krause et al., 2013), EuroSAT (Helber et al., 2019), FGVC-Aircraft (Maji et al., 2013), Caltech101 (Fei-Fei et al., 2004), DTD (Cimpoi et al., 2014), Flowers101 (Nilsback & Zisserman, 2008) and Oxford Pets (Parkhi et al., 2012). We use the "Final Acc" metric to represent the accuracy of the last round, while the "Average Acc" metric provides a comprehensive evaluation of PSP by computing the average accuracy of the final round across all datasets. All experiments were conducted three times, and the results are reported as average values.

**Implementation Details.** We adopt PCB (Bang et al., 2024) as the baseline model, following the setup of eight rounds (*i.e.*, $R = 8$). In each round, we select the query set whose size corresponds to the number of classes in the datasets, *i.e.*, $n_t^s = K$. For VSSP, coefficient $\beta$, hyperparameter $\tau$ and $\gamma$ are set to 0.7, 0.1 and 0.9, respectively. In UST, we apply the same data augmentation to the remaining unlabeled data five times, *i.e.* $L = 5$, which involves random resized crop to 224×224 with scale=(0.08, 1.0), random grayscale conversion with probability 0.2, color jittering, random horizontal flip, and normalization. All the experiments are conducted using the PyTorch platform and executed on NVIDIA RTX 3090 GPUs. More implementation details can be found in Appendix A.1.

Table 1: **Final accuracy on these commonly used downstream tasks using the ViT-B/32 image encoder.** The performances with the pre-trained zero-shot CLIP model are reported from (Rakesh & Jain, 2021). The performance with the entire labeled dataset during prompt learning is marked as "Fully Labeled Data", serves as the upper bound for comparison.

| Method | DTD | Oxford Pets | EuroSAT | Flowers102 | Caltech101 | Stanford Cars | Aircraft | Average Acc (↑) |
|---|---|---|---|---|---|---|---|---|
| CLIP (Zero-Shot) | 44.5 | 87.0 | 49.4 | 66.7 | 87.9 | 59.4 | 21.2 | 59.44 |
| Random | 58.77±1.94 | 78.30±0.74 | 77.62±1.12 | 92.92±0.61 | 89.55±1.00 | 65.96±0.08 | 30.69±0.30 | 70.54 |
| GCNAL (Caramalau et al., 2021) | 59.82±1.52 | 82.09±0.59 | 82.12±0.33 | 93.19±0.23 | 92.44±0.60 | 65.34±0.32 | 29.84±0.48 | 72.12 |
| ALFA-Mix (Parvaneh et al., 2022) | 61.28±0.41 | 83.13±0.13 | 82.39±0.93 | 96.76±0.17 | 95.37±0.11 | 71.04±0.67 | 27.83±0.25 | 74.01 |
| Entropy (Holub et al., 2008) | 59.18±1.31 | 76.81±1.38 | 75.46±3.39 | 94.80±0.75 | 91.67±0.09 | 66.68±0.91 | 25.80±0.78 | 70.06 |
| + AE | 60.80±1.18 | 78.35±1.30 | 79.97±2.70 | 96.06±0.63 | 92.87±0.20 | 65.99±0.26 | 26.69±1.34 | 71.53 |
| + AS | 59.34±0.81 | 79.88±1.43 | 79.88±0.43 | 95.67±1.19 | 93.28±0.55 | 68.54±0.09 | 26.04±1.27 | 71.75 |
| + PCB (Bang et al., 2024) | 59.73±1.96 | 80.44±1.24 | 80.02±2.88 | 96.16±0.45 | 92.41±0.50 | 67.18±0.28 | 26.78±0.87 | 71.93 |
| + PCB (AE) | 60.07±1.69 | 80.87±0.60 | 81.72±0.53 | 96.33±0.06 | 93.14±0.51 | 66.42±0.86 | 27.09±0.13 | 72.23 |
| + PCB (AS) | 59.50±1.99 | 80.94±1.05 | 80.75±1.15 | **96.94±0.19** | 93.48±0.26 | 68.93±0.86 | 27.58±0.43 | 72.59 |
| Coreset (Sener & Savarese, 2018) | 50.39±0.54 | 76.70±0.52 | 68.09±1.54 | 88.65±0.68 | 88.78±0.49 | 61.75±0.60 | 24.32±0.45 | 65.53 |
| + AE | 51.89±1.38 | 78.08±1.07 | 67.02±2.86 | 89.06±0.62 | 88.99±0.82 | 60.65±0.33 | 25.88±0.70 | 66.08 |
| + AS | 52.76±1.21 | 78.89±0.84 | 70.63±0.54 | 89.73±0.93 | 90.63±0.54 | 64.15±0.77 | 26.11±0.86 | 67.19 |
| + PCB (Bang et al., 2024) | 55.77±1.33 | 76.84±1.10 | 77.50±4.64 | 91.30±0.90 | 89.96±0.03 | 63.63±0.27 | 25.38±0.64 | 68.63 |
| + PCB (AE) | 57.09±0.63 | 78.60±1.14 | 79.28±1.40 | 91.70±0.29 | 90.29±0.30 | 62.08±0.35 | 26.19±1.40 | 69.31 |
| + PCB (AS) | 56.38±0.73 | 79.50±0.91 | 79.28±1.42 | 92.33±0.84 | 91.70±0.48 | 65.75±0.55 | 26.22±0.47 | 70.17 |
| BADGE (Ash et al., 2020) | 58.98±1.38 | 80.03±1.19 | 79.79±0.94 | 96.33±0.39 | 92.54±0.01 | 68.07±0.61 | 31.25±0.45 | 72.43 |
| + AE | 59.97±0.71 | 81.94±0.55 | 80.57±1.40 | 96.24±0.29 | 92.93±0.02 | 67.10±0.47 | 31.04±0.32 | 72.83 |
| + AS | 61.52±1.25 | 82.33±0.72 | 81.66±0.41 | 96.44±0.16 | 93.79±0.25 | 70.56±0.31 | 31.79±0.74 | 74.01 |
| + PCB (Bang et al., 2024) | 60.28±1.06 | 80.22±1.69 | 81.98±0.81 | 96.12±0.12 | 92.21 ± 0.92 | 68.50±0.26 | 31.35±0.21 | 72.95 |
| + PCB (AE) | 61.92±1.06 | 81.93±0.88 | 80.70±3.67 | 96.35±0.27 | 92.52±0.32 | 67.70±0.84 | 31.80±0.08 | 73.27 |
| + PCB (AS) | 62.33±1.06 | 83.16±0.18 | 81.50±1.11 | 96.71±0.29 | 93.85±0.37 | 70.70±0.79 | 32.27±0.66 | 74.36 |
| PSP | **65.66±0.88** | **86.57±0.93** | **85.43±0.08** | 96.35±0.57 | **93.87±0.31** | **73.84±0.29** | **36.42±0.45** | **76.87** |
| Fully Labeled Data | 74.7 | 89.3 | 94.5 | 97.9 | 94.4 | 80.8 | 43.4 | 82.14 |

### 4.2 OVERALL RESULTS

We evaluate our PSP against three PCB variants that implement different description augmentation: one that omits augmentation, one that uses the average score, and another that employs the average embedding, denoted as PCB, PCB (AS), and PCB (AE), respectively. The details and experimental results of description augmentation are provided in Appendix A.4. Moreover, we compare our PSP with the pre-trained zero-shot CLIP, ALFA-Mix, GCNAL and the Random approach that randomly selects a query set in each round. The results on downstream tasks with ViT-B/32 image encoder are summarized in Table 1. For smaller datasets, PSP achieves improvements of **3.33%**, **3.41%** and **4.15%**

Table 2: **Final accuracy with the ViT-B/32 CLIP image encoder on DTD.** The baseline model is combined with UST, and VSSP.

| Method | DTD | Oxford Pets | EuroSAT | Aircraft | Average |
|---|---|---|---|---|---|
| w/o VSSP | 64.36 | 85.91 | 84.63 | 34.14 | 67.26 |
| w/o UST | 63.77 | 85.55 | 83.97 | 32.40 | 66.42 |
| PSP | **65.66** | **86.57** | **85.43** | **36.42** | **68.52** |
| Full Labeled Data | 74.7 | 89.3 | 94.5 | 43.4 | 75.5 |

over PCB (AS) on DTD, Oxford Pets and Aircraft, respectively. For larger datasets, PSP demonstrates gains of **3.14**% and **3.93**% over PCB (AS) on Stanford Cars and EuroSAT, respectively. Furthermore, PSP achieves improvements of **2.51**% on the "Average Acc" metric across these datasets. These results provide convincing evidence that our PSP effectively enhances performance.

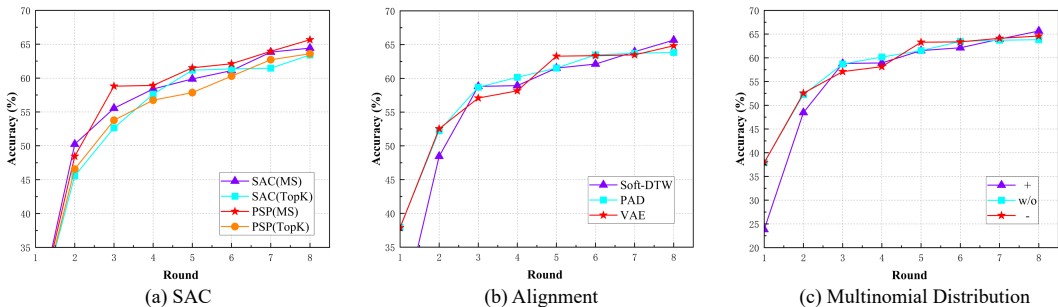

Figure 3: **Influence of different designs of the Vectorized Soft Actor-Critic Sampling Policy.** (a) Different query strategies (*i.e.*, MS and TopK) in VSSP. TopK means selecting the samples with the highest probabilities in the action. (b) Various alignment algorithms in VSSP. (c) Different usages of MS indicator within VSSP.

Table 3: **Ablation study on DTD, evaluating the impact of hyperparameter** $\beta$.

| $\beta$ | Final Accuracy | $\beta$ | Final Accuracy |
|---|---|---|---|
| 0.0 | 63.89 | 0.5 | 65.42 |
| 0.1 | 64.95 | **0.7** | **65.66** |
| 0.3 | 64.24 | 0.9 | 64.66 |

## 4.3 ABLATION STUDY

**Effectiveness of each component in PSP.** We present the influence of each component in PSP in Table 2, reporting the final accuracy on DTD, Oxford Pets, EuroSAT, and Aircraft. *w/o VSSP* indicates the removal of the sampling policy, which is equivalent to PCB combined with UST, while *w/o UST* denotes the absence of pseudo-labeled data, corresponding to PCB integrated with VSSP. We can notice that the average performance across four datasets reduced by **1.26** % and **2.10** % respectively when sampling policy and pseudo-labeled data are removed. These results suggest that both VSSP and UST are crucial for effectively guiding the student CLIP' prompt learning. Consequently, we conclude that guiding the sampling policy with prompts effectively enhances the optimization of the prompt template by considering both selected and unselected samples.

**Effectiveness of each parts in VSSP.** To study the effectiveness of different designs for VSSP, we conduct ablation studies on DTD and report the accuracy for each round. VSSP is built upon SAC with vectorized critics and real-pseudo hybrid reward. First, we remove the vectorized critics from VSSP, denoted as SAC (MS), and observe a substantial performance drop compared with PSP (MS), as detailed in Figure 3a. These results indicate that the vectorized critics play an indispensable role in achieving the performance gains. Moreover, we replace MS with TopK as the query strategy within VSSP, denoted as PSP (TopK). As shown in Figure 3a, PSP (MS) consistently outperforms PSP (TopK), validating that MS is more compatible with our PSP.

Second, we analyze the impact of different alignment methods (*i.e.*, Soft-DTW, PAD, and VAE) on DTD, as shown in Figure 3b. We adopt Soft-DTW as the default alignment algorithm, which improves the final accuracy by **1.83%** and **0.83%** compared to PAD and VAE, respectively.

Third, we remove the real-pseudo hybrid reward from VSSP (*i.e.*, removing the MS indicator in Equation 4, denoted as "w/o") and observe a **1.83%** performance decrease, as shown in Figure 3c. This finding demonstrates that real-pseudo hybrid reward has a significant impact on performance. Notably, the results show that using a positive coefficient improves the final accuracy by **1.06%** compared to the negative coefficient. This supports our analysis that since MS indicator is inherently negative, maximizing the reward leads to a smaller absolute sum of real and pseudo rewards, which correlates with better classification performance.

**Average accuracy in each round.** To more thoroughly analyze the performance of PSP in each round, we report the average accuracy across seven commonly used datasets, referred to as the learning curve. As shown in Figure 4, PSP consistently outperforms all three PCB variants after the initial round, with the performance disparity progressively increasing. It has been validated that PSP has almost achieved an enhancement in overall performance in each round compared to the three PCB variants.

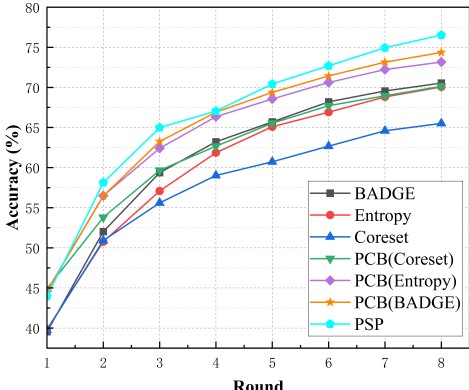

Figure 4: **Learning curve.** Average accuracy across downstream tasks with the ViT-B/32 image encoder for each round.

**Hyperparameter sensitivity.** To determine the optimal coefficient $\beta$, we compare PSP with different values of the hyperparameter $\beta$ in Table 3. The results show that PSP exhibits robustness with respect to the hyperparameter $\beta$. Ultimately, we select $\beta = 0.7$ as the default setting for superior performance. In Algorithm 1, the target value smoothing coefficient $\tau$ is used to stabilize the training of the V-Critic network. We conduct experiments comparing different values of the $\tau$ and conclude that PSP is insensitive to $\tau$, showing minimal performance variation. Additionally, PSP is robust to the discount factor $\gamma$. More ablation study can be found in Appendix A.2.

## 5  CONCLUSION

In this work, we propose a novel Prompt-Guided Self-Training Sampling Policy (PSP) for APL, which integrates SAC with a tailored real-pseudo hybrid reward and vectorized critics to leverage prompt in steering sample selection toward those that drive the optimization of prompt template, by jointly considering both selected and unselected samples. PSP constructs a self-training framework composed of VSSP and UST. VSSP utilizes learned prompts and image features to compute a real-pseudo hybrid reward, which is fed into vectorized critics to estimate the each sample's Q-value and compute gradients for actor's optimization. This process enables the actor guided by the prompt to refine its sampling policy in an End-to-End manner and identify the most crucial samples for the student CLIP's prompt learning, distinguishing it from PCB. UST extracts valuable complementary information from unselected samples by utilizing the teacher CLIP to generate reliable pseudo-labeled data based on uncertainty and confidence. Extensive experiments prove that PSP can identify the most crucial samples for prompt learning to maximize performance within a constrained budget.

ACKNOWLEDGMENT

This work was supported by National Natural Science Foundation of China [U24A20277, 62225207, 62476260, 62436008, 92267205, 62503466, 92467301, 92367301], Natural Science Foundation of Liaoning Province [2024-MSBA-83, 2025-MS-085], the Fundamental Research Funds for the Central Universities under Grant WK2100000057, the National Program for Funded Postdoctoral Researchers [GZB20230805], Fundamental Research Project of SIA,[2024JC1K10, 2024JC3K03], the State Key Laboratory of Robotics and Intelligent Systems of China [2025-Z12].

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

# A APPENDIX

## A.1 EXPERIMENTAL DETAILS

**Dataset.** Stanford Cars (Krause et al., 2013) is a complex dataset known for its fine-grained categorization. It contains 16,185 images spanning 196 car models. EuroSAT (Helber et al., 2019) comprises 27,000 Sentinel-2 satellite images grouped into 10 classes of land use and land cover, making it a valuable resource for remote sensing. FGVC-Aircraft (Maji et al., 2013) consists of 10,200 images representing 102 distinct aircraft model variants. Caltech101 (Fei-Fei et al., 2004) includes 9,146 images, divided into 101 object categories, plus an additional background category. DTD (Cimpoi et al., 2014) features 5,640 images across 47 texture categories, offering a diverse selection of texture patterns sourced from natural environments. Flowers101 (Nilsback & Zisserman, 2008) contains 8,189 images categorized into 102 flower species, exhibiting considerable intra-class variation and inter-class similarities. Oxford Pets (Parkhi et al., 2012) consists of 7,349 images of cats and dogs, covering 37 different breeds.

**Experimental details.** In Algorithm 1, the learning rate $\lambda_V$, $\lambda_Q$ and $\lambda_\pi$ are all set to 3e-4. To ensure a fair comparison, we adopt ViT-B/32 as the default backbone for the student CLIP, the teacher CLIP in all experiments. Throughout all rounds, the prompt learning process for the student CLIP is optimized with the cross-entropy loss using SGD at a learning rate of 0.002, a batch size of 32, and 200 epochs across all datasets. For the text prompt, we adopt AS to realize the description augmentation for enhancing performance and set the size of the learnable tokens $M$ to 16.

## A.2 ADDITIONAL RESULTS

**Analysis of the state modeling in VSSP.** We are intend to incorporate the classification score information of unlabeled samples to construct gradient embeddings as state in Equation 2, thereby enriching the state representation and facilitating better learning of the sampling policy. To verify the effectiveness of the classification score information, we conduct experiments where only the features of unlabeled samples are used as the state. The results indicate that using only features as the state in PSP is suboptimal, as the accuracy decreases from **65.66%** to **63.36%** when compared with using gradient embeddings. We can conclude that modeling the state with classification score information helps refine the sampling policy.

**Further analysis of UST.** To further analyze the effectiveness of UST, we conduct experiments on DTD and EuroSAT to compare the UST module with representative semi-supervised (Chakraborty et al., 2024) and unsupervised (Huang et al., 2022) prompt learning methods, as shown in Table 4. We include the following methods for comparison: UPL (Huang et al., 2022), which uses CLIP with a ResNet-50 backbone for both pseudo-labeling and inference; UPL* (Huang et al., 2022), an enhanced version of UPL that leverages multiple CLIP backbones (ResNet-101, ViT-B/32, ViT-B/16, and ViT-L/14) for improved pseudo-labeling, while retaining CLIP with ResNet-50 backbone for inference; and XPL (Chakraborty et al., 2024), a semi-supervised prompt learning method that uses the same number of labeled samples as UST. Experimental results indicate that UST outperforms UPL, UPL*, and XPL on both DTD and EuroSAT, demonstrating that filtering reliable pseudo-labeled data through UST effectively enhances prompt learning on downstream tasks.

**Analysis of the accuracy and number of reliable pseudo-labeled data.** To further study the quality of pseudo-labeled data, we report the accuracy and the number of reliable pseudo-labeled data on the DTD dataset. As shown in Table 5, both metrics steadily increase as the rounds progress, with the accuracy approaching **94%** in the final round. These results strongly demonstrate that UST effectively filters out reliable pseudo labels, and that the high accuracy of the pseudo-labeled data substantially mitigates the negative impact of incorrect pseudo labels.

**Performance comparison with classical prompt learning methods.** We evaluate the proposed PSP against classical prompt learning methods, including CoOp (Zhou et al., 2022b) and CoCoOp (Zhou et al., 2022a), using the same number of real-labeled training samples as our approach. As illustrated in Table 8, PSP significantly outperforms CoOp and CoCoOp in terms of final accuracy across all datasets except Oxford Pets and Caltech101. Meanwhile, PSP achieves average accuracy improvements of **7.76%** over CoOp and **7.38%** over CoCoOp across these datasets. These results demonstrate that PSP effectively facilitates prompt optimization by annotating more valuable samples and extracting valuable complementary information from the remaining unlabeled data.

Table 4: **Ablation study with semi-supervised and unsupervised prompt learning methods.** We present the final accuracy on DTD and EuroSAT using ViT-B/32 as the image encoder for performance comparison with UPL and XPL.

| Method | DTD | EuroSAT |
|---|---|---|
| UPL (Huang et al., 2022) | 46.10 | 52.17 |
| UPL* (Huang et al., 2022) | 55.08 | 71.04 |
| XPL (Chakraborty et al., 2024) | 62.29 | 79.30 |
| UST | **62.65** | **81.59** |

Table 5: **Analysis of the accuracy and the number of reliable pseudo-labeled data on the DTD dataset in each round.**

| Metric | 1 | 2 | 3 | 4 | 5 | 6 | 7 | 8 |
|---|---|---|---|---|---|---|---|---|
| **Accuracy** | 45.47 | 47.99 | 83.30 | 90.67 | 91.35 | 92.06 | 93.25 | 93.93 |
| **Number** | 436 | 455 | 494 | 522 | 534 | 532 | 543 | 559 |

**Analysis of the versatility of PSP for different VLMs.** To further prove the versatility of PSP for different VLMs, we replace CLIP with SigLIP (Zhai et al., 2023). As shown in the Table 6, PSP significantly enhances the prompt learning of SigLIP compared to the baseline PCB. Therefore, we can draw a conclusion that the proposed method offers a feasible solution for applying standard VLMs to a wider range of domains under limited labeling budget.

**Analysis of the behavior of the learned sampling policy.** To evaluate the behavior of our sampling policy, we compare it with BADGE, Entropy, and Coreset, and report the overlap ratio, computed as the intersection between the samples selected by both methods divided by the number of samples selected by our sampling policy. The results in Table 7 show that PSP exhibits relatively high consistency with Coreset in the early rounds (Rounds 2-4), while in later rounds (Rounds 5-8), its overlap with Entropy increases. This indicates that the sampling policy initially favors diverse samples to establish broad coverage, and gradually shifts to uncertain samples as the model's classification capability improves.

Notably, since the overlap between PSP and BADGE is not consistently higher than that with the other heuristics, we can conclude that PSP is not merely imitating a fixed hybrid rule such as BADGE. Instead, it continuously interacts with the prompt learning process and adaptively adjusts its sampling policy over rounds.

**Analysis of model efficiency.** We evaluate the model efficiency of PSP against various PCB variants under the same backbone (ViT-B/32) by reporting training time in each round in seconds (*e.g.*, 305, 380...) and the total training time in hours, as illustrated in Table 9. Compared to PCB combined with BADGE, PSP requires less training time while achieving a **3.33%** higher final accuracy. The experimental results demonstrate that PSP strikes a balance between model efficiency and performance improvement.

**Influence of various image encoder types.** To explore whether different image encoder types affect the performance of PSP, we report the results on various image encoder backbones (*i.e.*, RN50, RN101, and ViT-B/16) in Table 10. It is demonstrated that PSP almost achieves performance improvements compared to other methods regardless of the image encoder backbone used. Notably, PSP achieves the highest performance on various image encoder backbones on the "Average Acc" metric, strongly proving that PSP comprehensively outperforms three PCB variants across these commonly used datasets.

**Influence of various buffer thresholds.** To determine the optimal buffer threshold, we compare different values in Figure 5. The results indicate that setting the buffer threshold $\tau_b$ to 1 achieves the highest performance. Given the limited number of experiences, we sample one experience per gradient step for updates to fully utilize each entry in the buffer.

**Performance comparison on ImageNet dataset.** To evaluate our method on more advanced and large-scale datasets, we conduct performance comparison on ImageNet (Deng et al., 2009) dataset. As shown in Figure 6a, PSP achieves comparable accuracy to PCB in the first two rounds but gradually outperforms it in later rounds, ultimately reaching the highest performance of **68.41%** on the final accuracy on the large-scale ImageNet dataset. These results provide powerful evidence that PSP effectively reduces the reliance on large-scale labeled datasets in prompt learning.

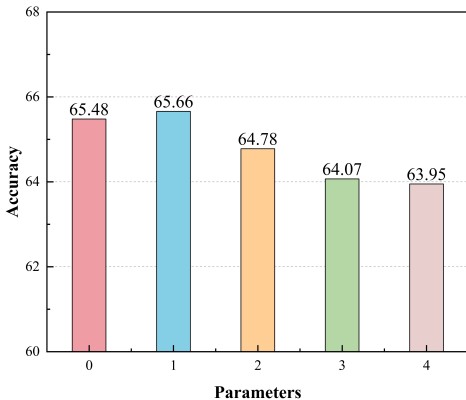

Figure 5: **Buffer threshold analysis on DTD.** We report the final accuracy for different buffer thresholds.

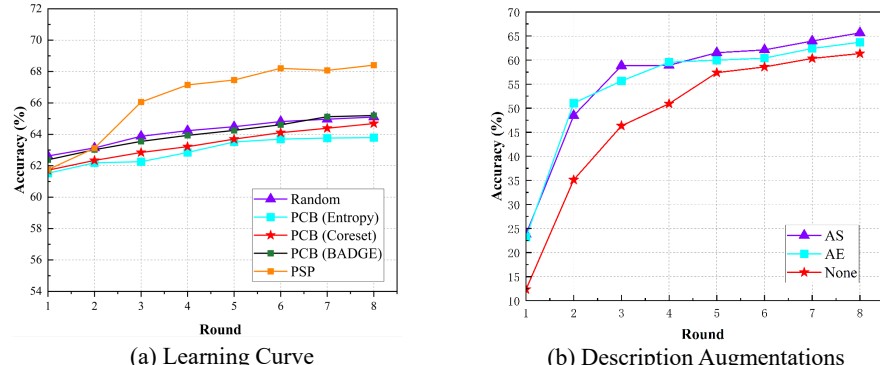

(a) Learning Curve          (b) Description Augmentations

Figure 6: **(a) Learning curve on ImageNet dataset.** Classification accuracy on the ImageNet dataset using the ViT-B/32 image encoder at each round. **(b) Ablation study of various description augmentations (*i.e.*, AS, AE, and None) on DTD.** We report learning curves to evaluate the effectiveness of various description augmentations.

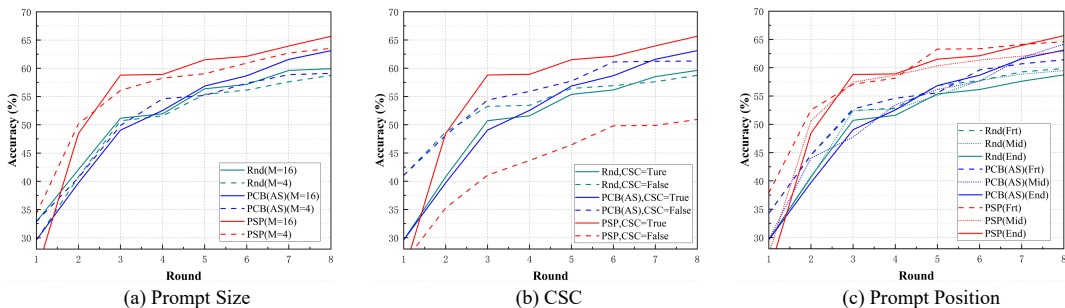

(a) Prompt Size          (b) CSC          (c) Prompt Position

Figure 7: **Learnable prompt analysis on DTD dataset.** We report accuracy for different prompt sizes, whether class-specific tokens are used, and varying prompt positions in each round.

Table 6: **Analysis of the versatility of PSP for different Vision-Language Models.**

| Method | DTD | Oxford Pets | EuroSAT | Aircraft | Average Acc |
|---|---|---|---|---|---|
| PCB (SigLIP) | 55.73 | 47.12 | 69.46 | 22.98 | 48.82 |
| PSP (SigLIP) | 59.75 | 62.99 | 86.18 | 28.65 | 59.39 |

Table 7: **Analysis of the behavior of the learned sampling policy.**

| Round | PSP vs CoreSet | PSP vs Entropy | PSP vs BADGE |
|---|---|---|---|
| 1 | 0.0113 | 0.0113 | 0.0113 |
| 2 | 0.1107 | 0.0959 | 0.0923 |
| 3 | 0.0920 | 0.0920 | 0.0958 |
| 4 | 0.1103 | 0.0608 | 0.0837 |
| 5 | 0.1094 | 0.1283 | 0.1283 |
| 6 | 0.1301 | 0.0892 | 0.0706 |
| 7 | 0.0784 | 0.1007 | 0.1045 |
| 8 | 0.0833 | 0.1023 | 0.1023 |

**Influence of description augmentation.** To find the optimal description augmentation suitable for PSP, we analyze the impact of different description augmentations (*i.e.*, AS, AE, and None) on DTD, and the results are presented in Figure 6b. It is confirmed that PSP utilizing AS as the description augmentation significantly outperforms the cases that use AE and no augmentation. Hence, we choose AS as the default description augmentation.

**Influence of increasing query size.** In Table 1, it is worth mentioning that the pre-trained zero-shot CLIP model outperforms PSP on Oxford Pets. To further analyze this anomalous phenomenon, we increase the query size $n_s$ from $K$ to $2K$ in each round and report the results of PSP on Oxford Pets in Table 11. We observe that the "Final Acc" metric increases as the size of the query set grows across all backbones, and PSP surpasses the performance of the pre-trained zero-shot CLIP model on various backbones when the query size reaches $n_s = 2K$.

**Learnable prompt analysis.** To further analyze which factors of the prompt affect the performance of PSP, we conduct experiments about several variables according to the learnable prompt, *e.g.* the size of learnable tokens (*i.e.*, prompt size) $M$, whether class-wise different tokens are allowed (CSC = True or False), and the position of class token (Front, Middle, and End). For prompt size $M$, the results are shown in Figure 7a. It is confirmed that PSP with a larger $M$ alternates in outperforming PSP with a smaller $M$ until the third round, after which it consistently maintains its performance lead through the final round. PSP with a larger $M$ achieves a **2.13**% improvement on the "Final Acc" metric compared to PSP with a smaller $M$, demonstrating that PSP is sensitive to the prompt size, with larger $M$ values yielding significantly better performance.

For class-wise different tokens, we analyze the performance gap between using the shared context vectors for all classes (*i.e.*, CSC = False) and using different context vectors for each class (*i.e.*, CSC = True), and the results shown in Figure 7b. The results show that the initial accuracy when CSC = True is lower than when CSC = False, but it eventually surpasses that of CSC = False in the second round, and then maintains its leading performance. It can be concluded that using class-wise different tokens offers a slight benefit to PSP. Additionally, during the first round, CSC = True is more susceptible to overfitting compared to CSC = False.

Ultimately, we conduct experiments to investigate the impact of the class token's position (*i.e.*, Front (Frt), Middle (Mid), and End) on active prompt learning for PSP, and the results reported in Figure 7c. The results reveal that the learning curves of PSP alternate in leading performance across different class token positions. PSP with the End position of class token slightly surpasses the others on the "Final Acc" metric. Therefore, we choose to place the class token after the context vectors for all experiments. It is demonstrated that the position of the class token has minimal impact on PSP in active prompt learning.

## A.3 PRELIMINARY

**Active Learning** identifies criteria for selecting the most informative samples under a limited labeling budget. Active learning methods are primarily applied in three distinct scenarios: membership query synthesis (Mahapatra et al., 2018; Mayer & Timofte, 2020), stream-based (Narr et al., 2016; Fang et al., 2017; Woodward & Finn, 2017), and recently most mainstream pool-based (Vijayanarasimhan & Grauman, 2011; Gosselin & Cord, 2008; Yang et al., 2015; Kapoor et al., 2010; Bang et al., 2024) setting. In this work, we follow the pool-based setting and define an unlabeled data pool from

Table 8: **Ablation study with classical prompt learning methods like CoOp and CoCoOp.** We report the final accuracy across seven datasets for a comprehensive comparison with CoOp and CoCoOp.

| Method | DTD | Oxford Pets | EuroSAT | Flowers102 | Caltech101 | Stanford Cars | Aircraft | Average Acc |
|---|---|---|---|---|---|---|---|---|
| CoOp (Zhou et al., 2022b) | 58.65 | 87.71 | 68.73 | 88.27 | 90.14 | 65.25 | 24.99 | 69.11 |
| CoCoOp (Zhou et al., 2022a) | 57.80 | **89.81** | 72.81 | 84.00 | **94.44** | 65.63 | 21.93 | 69.49 |
| PSP | **65.66** | 86.57 | **85.43** | **96.35** | 93.87 | **73.84** | **36.42** | **76.87** |

Table 9: **Analysis of efficiency on DTD.** All models are trained on a single RTX 3090 GPU with a batch size of 32.

| Method | 1 | 2 | 3 | 4 | 5 | 6 | 7 | 8 | Training Time (h) | Final Acc (%) |
|---|---|---|---|---|---|---|---|---|---|---|
| **Random** | 305 | 380 | 456 | 475 | 540 | 586 | 645 | 670 | 1.13 | 58.77 |
| **PCB(Entropy)** | 300 | 382 | 413 | 433 | 502 | 545 | 610 | 635 | 1.06 | 59.18 |
| **PCB(Coreset)** | 327 | 415 | 486 | 494 | 561 | 611 | 662 | 676 | 1.18 | 56.38 |
| **PCB(BADGE)** | 301 | 944 | 906 | 919 | 971 | 998 | 980 | 991 | 1.95 | 62.33 |
| **PSP** | 725 | 817 | 957 | 898 | 869 | 861 | 847 | 895 | 1.91 | 65.66 |

which a subset of samples is actively selected for annotation. Based on the criteria, active learning methods can be categorized into three main approaches: Uncertainty-based sampling (Gal et al., 2017; Wang et al., 2019), Diversity-based sampling (Hacohen et al., 2022; Shui et al., 2020), Hybrid sampling (Ash et al., 2020; Parvaneh et al., 2022; Caramalau et al., 2021), and RL-based sampling (Ash et al., 2020; Kirsch et al., 2019). **Uncertainty-based sampling**, a straightforward and effective strategy, focuses on selecting samples that the model struggles to learn, employing techniques such as Monte-Carlo Dropout (Gal et al., 2017; Kirsch et al., 2019), Entropy (Holub et al., 2008), and Least Confident (Lewis & Catlett, 1994). Holub *et al.* proposed Entropy (Holub et al., 2008) for object recognition, which selects the samples with the highest entropy for annotation. **Diversity-based sampling** focuses on selecting samples that represent the full data distribution to ensure diversity in the labeled data, including clustering (Hu et al., 2021) and Coreset (Sener & Savarese, 2018). Sener *et al.* introduced the elegant and mathematically rigorous Coreset (Sener & Savarese, 2018), which provides an approximate upper bound on the loss for feature space coverage-based active learning algorithms. **Hybrid sampling** (Ash et al., 2020; Parvaneh et al., 2022) takes into account both diversity and uncertainty, aiming to mitigate the issue of redundancy in Uncertainty-based sampling and the limitations of Diversity-based sampling, where basic feature coverage strategies may fall short in assessing the model's confidence in its predictions. ALFA-Mix (Parvaneh et al., 2022) utilizes unlabeled data to support active learning by interpolating between the representations of labeled and unlabeled instances and identifying features the model fails to recognize through inconsistencies in predicted labels. However, Hybrid sampling relies on fixed rules to balance diversity and uncertainty, limiting its adaptability across tasks. **RL-based sampling** formulates a sample selection policy, where Reinforcement Learning (RL) is applied to learn a policy that maximizes cumulative reward by selecting samples. Woodward et al. developed AOL (Woodward & Finn, 2017) that combines meta-learning and reinforcement learning for one-shot classification tasks. Liu et al. introduced DRAL (Liu et al., 2019) to guide an agent in acquiring pairwise annotated data. Notably, PAL (Fang et al., 2017) builds a deep Q-network as an adaptive policy for sample selection. Therefore, we believe that RL-based methods have the potential to incorporate prompts for guiding sample selection. However, AOL (Woodward & Finn, 2017) and PAL (Fang et al., 2017) model the decision of whether to annotate a streaming unlabeled sample as a binary classification problem, while MedSelect (Vrabac et al., 2022) and DARL (Liu et al., 2019) rely on pairwise data, making them unsuitable for direct application in Active Prompt Learning (APL). Therefore, we introduce Soft Actor-Critic (SAC) (Haarnoja et al., 2018), a representative reinforcement learning algorithm known for its robustness to hyperparameters and strong performance in continuous action spaces. By designing a customized real-pseudo hybrid reward and vectorized critics, SAC can be seamlessly integrated into APL.

### A.4 DESCRIPTION AUGMENTATION

Here, the new text prompt is converted below:

$$\boldsymbol{p}_{i,k} = [\boldsymbol{c}]_1[\boldsymbol{c}]_2\dots[\boldsymbol{c}]_M[\text{cls}_i][\text{which}][\text{is}][d_i^k] \tag{14}$$

where $d_i^k$ denotes the $k$-th description for class $i$, $\Delta_k = \{d_i^k\}_{k=1}^{\epsilon_i}$ represents $\epsilon_i$ descriptions for class $i$. Given the new text prompt, two possible prediction probabilities after description augmentation are expressed below:

Table 10: **Final accuracy of different image encoder architectures, including ResNet-50/101 and ViT-B/16.**

| | Method | DTD | Oxford Pets | EuroSAT | Flowers102 | Caltech101 | Stanford Cars | Aircraft | Average Acc (↑) |
|---|---|---|---|---|---|---|---|---|---|
| **RN50** | CLIP (Zero-Shot) | 44.7 | 85.4 | 41.1 | 65.9 | 82.1 | 55.8 | 19.3 | 55.9 |
| | Random | 56.62±0.97 | 74.65±0.50 | 79.10±2.31 | 92.06±0.54 | 84.11±0.75 | 61.34±0.57 | 29.15±0.32 | 68.18 |
| | GCNAL (Caramalau et al., 2021) | 55.26±0.51 | 78.24±1.19 | 80.92±0.54 | 92.92±0.60 | 88.00±0.44 | 64.31±0.76 | 28.23±0.59 | 69.70 |
| | ALFA-Mix (Parvaneh et al., 2022) | 59.02±0.62 | 82.04±0.61 | 82.37±0.17 | 95.22±0.34 | 91.62±0.36 | 67.36±0.09 | 30.45±0.27 | 72.58 |
| | Entropy (Holub et al., 2008) | 57.62±2.13 | 72.74±0.97 | 75.73±4.28 | 95.19±0.09 | 88.21±0.42 | 61.32±0.80 | 25.13±0.96 | 67.99 |
| | + PCB (Bang et al., 2024) | 56.44±0.39 | 75.49±0.45 | 81.69±1.63 | 95.30±0.59 | 88.78±0.43 | 62.02±0.17 | 25.75±0.35 | 69.35 |
| | + PCB (AE) | 59.02±0.59 | 76.59±0.12 | 81.77±1.51 | 95.75±0.23 | 89.41±0.53 | 61.05±0.99 | 26.44±0.81 | 70.00 |
| | + PCB (AS) | 59.34±1.09 | 78.59±1.41 | 83.26±0.35 | 96.17±0.27 | 90.49±0.02 | 63.52±0.31 | 26.46±0.99 | 71.12 |
| | Coreset (Sener & Savarese, 2018) | 48.74±1.00 | 69.87±2.36 | 70.02±4.16 | 85.02±1.51 | 83.34±1.33 | 57.93±0.56 | 25.38±0.62 | 62.90 |
| | + PCB (Bang et al., 2024) | 51.63±0.30 | 71.15±1.64 | 77.74±2.13 | 88.79±0.98 | 85.54±0.84 | 58.67±0.37 | 25.33±0.63 | 65.64 |
| | + PCB (AE) | 51.69±1.25 | 73.70±0.27 | 77.74±3.33 | 89.27±1.69 | 86.69±0.57 | 57.63±0.55 | 25.17±0.37 | 65.98 |
| | + PCB (AS) | 53.15±1.37 | 75.53±1.64 | 79.79±1.06 | 89.50±1.39 | 87.15±1.44 | 60.61±0.54 | 25.88±0.10 | 67.37 |
| | BADGE (Ash et al., 2020) | 58.35±1.20 | 75.06±0.50 | 80.94±0.55 | 95.56±0.54 | 89.67±0.30 | 63.96±0.53 | 28.12±1.03 | 70.24 |
| | + PCB (Bang et al., 2024) | 57.41±0.17 | 76.51±1.83 | 80.06±0.97 | 95.66±0.28 | 89.06±0.21 | 63.18±0.77 | 29.23±0.35 | 70.16 |
| | + PCB (AE) | 59.20±1.25 | 76.77±0.65 | 81.96±0.60 | 95.72±0.31 | 89.57±0.19 | 62.62±0.26 | 28.85±1.59 | 70.67 |
| | + PCB (AS) | 59.14±1.08 | 80.09±0.85 | 81.60±2.89 | 96.18±0.07 | 90.76±0.34 | 66.20±0.69 | 29.61±0.78 | 71.94 |
| | PSP | **62.71±0.93** | **85.45±0.91** | **87.74±0.24** | 95.37±0.74 | 90.70±0.45 | **67.53±0.41** | **32.37±0.63** | **74.55** |
| | Fully Labeled Data | 71.6 | 88.0 | 93.6 | 97.6 | 92.8 | 78.8 | 42.6 | 80.71 |
| **RN101** | CLIP (Zero-Shot) | 43.9 | 86.2 | 33.1 | 65.7 | 85.1 | 62.3 | 19.5 | 56.54 |
| | Random | 58.29±1.24 | 79.08±1.39 | 77.21±4.13 | 92.87±0.43 | 87.55±0.75 | 70.02±0.36 | 32.76±0.29 | 71.11 |
| | GCNAL (Caramalau et al., 2021) | 57.00±0.54 | 82.04±0.49 | 81.68±0.69 | 91.66±0.13 | 90.55±0.23 | 68.99±0.24 | 31.18±0.37 | 71.87 |
| | ALFA-Mix (Parvaneh et al., 2022) | 61.33±0.33 | 85.04±0.48 | 82.91±0.49 | 96.82±0.34 | 92.82±0.25 | 75.20±0.06 | 31.54±0.45 | 75.09 |
| | Entropy (Holub et al., 2008) | 57.17±1.54 | 78.63±0.99 | 74.88±1.26 | 96.26±0.11 | 91.02±0.48 | 70.09±0.16 | 27.49±0.69 | 70.79 |
| | + PCB (Bang et al., 2024) | 58.81±1.39 | 80.14±1.27 | 79.91±2.06 | 96.26±0.25 | 91.62±0.30 | 70.87±0.45 | 28.11±0.37 | 72.25 |
| | + PCB (AE) | 59.81±1.34 | 82.65±0.99 | 81.23±1.26 | 96.47±0.39 | 92.16±0.90 | 70.14±0.56 | 27.96±1.63 | 72.92 |
| | + PCB (AS) | 60.70±1.09 | 83.64±1.02 | 82.43±1.35 | 96.49±0.17 | 92.87±0.20 | 73.62±0.67 | 28.68±0.83 | 74.06 |
| | Coreset (Sener & Savarese, 2018) | 52.23±1.76 | 74.02±1.81 | 66.62±0.54 | 87.90±0.92 | 87.23±1.18 | 65.83±0.43 | 26.37±0.42 | 65.74 |
| | + PCB (Bang et al., 2024) | 54.75±2.93 | 76.43±1.61 | 75.39±1.94 | 91.08±0.37 | 89.36±0.20 | 66.97±0.75 | 27.28±0.33 | 68.75 |
| | + PCB (AE) | 56.38±1.55 | 77.11±1.86 | 76.99±0.65 | 91.61±1.30 | 89.90±0.06 | 65.38±0.62 | 27.72±0.39 | 69.30 |
| | + PCB (AS) | 57.31±2.07 | 81.14±0.24 | 78.49±1.99 | 91.80±0.28 | 90.11±0.30 | 69.11±0.73 | 28.31±0.78 | 70.90 |
| | BADGE (Ash et al., 2020) | 59.93±1.25 | 80.77±1.31 | 78.23±2.22 | 96.26±0.07 | 91.35±0.32 | 71.43±0.97 | 32.56±0.64 | 72.93 |
| | + PCB (Bang et al., 2024) | 60.20±1.89 | 80.94±0.42 | 79.55±1.37 | 95.79±0.38 | 91.75±0.44 | 71.35±0.39 | 32.62±1.48 | 73.17 |
| | + PCB (AE) | 62.59±0.84 | 83.02±0.89 | 81.50±0.69 | 96.49±0.26 | 92.51±0.32 | 71.42±0.77 | 32.76±0.76 | 74.33 |
| | + PCB (AS) | 62.17±1.04 | 83.48±2.13 | 81.14±1.57 | 96.47±0.18 | 92.87±0.18 | 74.04±0.39 | 32.84±0.85 | 75.43 |
| | PSP | **63.95±0.74** | **87.43±0.61** | **87.19±0.19** | 96.10±0.51 | 92.41±0.42 | **75.28±0.44** | **37.98±0.37** | **77.19** |
| | Fully Labeled Data | 74.2 | 91.1 | 92.9 | 97.8 | 94.7 | 83.7 | 46.0 | 82.91 |
| **ViT-B/16** | CLIP (Zero-Shot) | 46.0 | 88.9 | 54.1 | 70.4 | 88.9 | 65.6 | 27.1 | 63.0 |
| | Random | 62.63±1.81 | 84.36±1.34 | 81.14±1.83 | 94.98±0.06 | 90.95±0.85 | 73.62±0.30 | 38.88±0.25 | 75.22 |
| | GCNAL (Caramalau et al., 2021) | 62.58±0.65 | 90.23±1.68 | 82.98±0.57 | 95.37±0.59 | 93.70±0.05 | 73.25±0.21 | 38.03±0.31 | 76.59 |
| | ALFA-Mix (Parvaneh et al., 2022) | 66.38±0.26 | 89.81±0.43 | 84.38±0.26 | 98.15±0.18 | 95.24±0.32 | 79.12±0.19 | 39.55±0.89 | 78.95 |
| | Entropy (Holub et al., 2008) | 62.49±0.39 | 82.56±0.49 | 77.93±0.90 | 97.63±0.42 | 93.04±0.41 | 74.35±0.59 | 33.27±0.72 | 74.47 |
| | + PCB (Bang et al., 2024) | 64.93±1.02 | 84.89±0.59 | 83.48±1.37 | 97.75±0.08 | 94.23±0.23 | 75.68±0.26 | 36.03±0.43 | 76.71 |
| | + PCB (AE) | 64.36±0.47 | 87.08±0.90 | 83.55±1.95 | 98.06±0.35 | 94.56±0.34 | 75.15±0.55 | 36.60±1.58 | 76.91 |
| | + PCB (AS) | 63.81±1.24 | 88.03±0.60 | 85.92±0.85 | **98.48±0.14** | 94.89±0.28 | 77.58±0.43 | 35.84±1.71 | 77.79 |
| | Coreset (Sener & Savarese, 2018) | 56.07±0.90 | 82.17±1.82 | 72.17±2.72 | 92.12±1.45 | 90.66±0.45 | 70.12±0.83 | 33.28±0.45 | 70.94 |
| | + PCB (Bang et al., 2024) | 59.07±0.63 | 83.09±1.19 | 80.25±3.12 | 94.79±0.31 | 90.60±0.80 | 71.27±0.19 | 34.06±0.66 | 73.30 |
| | + PCB (AE) | 60.54±0.86 | 84.52±0.23 | 84.04±2.92 | 94.94±0.55 | 92.15±0.09 | 70.10±1.03 | 33.36±0.03 | 74.24 |
| | + PCB (AS) | 61.98±1.04 | 86.77±0.69 | 83.85±2.45 | 95.44±0.82 | 92.97±0.29 | 72.96±0.63 | 35.24±0.49 | 75.60 |
| | BADGE (Ash et al., 2020) | 62.84±2.17 | 85.54±1.30 | 82.22±1.94 | 97.97±0.41 | 93.77±0.51 | 76.55±0.78 | 39.64±0.14 | 76.93 |
| | + PCB (Bang et al., 2024) | 64.89±1.45 | 86.22±0.71 | 81.53±3.11 | 98.32±0.21 | 93.75±0.28 | 76.36±0.27 | 40.20±0.30 | 77.32 |
| | + PCB (AE) | 65.25±1.28 | 87.23±0.35 | 84.04±2.92 | 98.21±0.29 | 94.51±0.44 | 75.84±0.21 | 39.93±0.21 | 77.86 |
| | + PCB (AS) | 64.95±1.47 | 88.10±1.49 | 83.85±2.45 | 98.19±0.17 | **95.12±0.26** | 78.19±0.48 | 40.56±0.51 | 78.42 |
| | PSP | **67.32±0.83** | **89.67±0.78** | **89.22±0.16** | 96.95±0.44 | 95.10±0.39 | **82.06±0.37** | **43.50±0.55** | **80.54** |
| | Fully Labeled Data | 77.7 | 92.7 | 95.1 | 99.0 | 95.3 | 85.3 | 53.6 | 85.53 |

Table 11: **Ablation study on Oxford Pets, which evaluates the impact of increasing query size $n_s$.**

| Method | $n_s$ | RN50 | RN101 | ViT-B/32 | ViT-B/16 |
|---|---|---|---|---|---|
| CLIP (Zero-Shot) | 0 | 85.4 | 86.2 | 87.0 | 88.9 |
| Random | 37 | 74.65 | 79.08 | 78.30 | 84.36 |
| PSP | 37 | 85.45 | 87.43 | 86.57 | 89.67 |
| PSP | 74 | 86.40 | 89.34 | 88.72 | 91.44 |
| Full Labeled Data | - | 88.0 | 91.1 | 89.3 | 92.7 |

(1) Average Similarity (AS):

$$p(y = i \mid x) = \frac{1}{\epsilon_i} \sum_{j=1}^{\epsilon_i} p(y = i \mid x, d_i^j) \tag{15}$$

$$p(y = i \mid x, d_i^j) = \frac{\exp(\cos(\boldsymbol{f}_V^s, \boldsymbol{f}_{T,i,j}^s)/\omega)}{\sum_{i=1}^{K} \sum_{j=1}^{\epsilon_i} \exp(\cos(\boldsymbol{f}_V^s, \boldsymbol{f}_{T,i,j}^s)/\omega)} \tag{16}$$

where $\boldsymbol{f}_{T,i,j}^s = \mathcal{F}_T^s(\boldsymbol{p}_{i,j}^s)$ denotes text feature corresponding to description $d_i^j$. $K$ represents the number of classes in a downstream task, and $\omega$ indicates a temperature scaling parameter.

(2) Average Embedding (AE):

$$p(y = i \mid x) = \frac{\exp(\cos(\boldsymbol{f}_V^s, \boldsymbol{f}_{T,i}^s)/\omega)}{\sum_{i=1}^{K} \exp(\cos(\boldsymbol{f}_V^s, \boldsymbol{f}_{T,i}^s)/\omega)} \tag{17}$$

$$\boldsymbol{f}_{T,i}^s = \frac{1}{\epsilon_i} \sum_{j=1}^{\epsilon_i} \boldsymbol{f}_{T,i,j}^s \tag{18}$$

The main difference between the two probability scores is that AS computes the cosine similarity for each text feature before averaging, whereas AE averages the text features first and then computes the similarity.

## A.5 SOFT DYNAMIC TIME WARPING

Given two vector sequences of unequal lengths: $\boldsymbol{Q}_\theta = [Q_\theta^{(1)}, Q_\theta^{(2)}, \ldots, Q_\theta^{(n_t^u)}]$ and $\hat{\boldsymbol{Q}} = [\hat{Q}^{(1)}, \hat{Q}^{(2)}, \ldots, \hat{Q}^{(n_{t+1}^u)}]$ **Compute pairwise distance matrix $\mathbf{D} \in \mathbb{R}^{n_t^u \times n_{t+1}^u}$:**

$$D_{i,j} = \|Q_\theta^{(i)} - \hat{Q}^{(j)}\|^2 \tag{19}$$

**Accumulated cost matrix $R$.** The accumulated cost matrix is initialized with $R_{0,0}$, followed by dynamic programming computation incorporating the soft minimum as described below.

$$R_{i,j} = D_{i,j} + \min_\gamma \left\{ R_{i-1,j}, R_{i,j-1}, R_{i-1,j-1} \right\}$$
$$\min_\gamma(a, b, c) = -\gamma \log \left( e^{-a/\gamma} + e^{-b/\gamma} + e^{-c/\gamma} \right) \tag{20}$$

**Path extraction.** The alignment path is derived by identifying the minimum cumulative cost path within the accumulated cost matrix.

$$\pi = \left\{ (i_1, j_1), (i_2, j_2), \ldots, (i_T, j_T) \right\} \tag{21}$$

where the integer $i_1 \in [1, n_t^u]$ indicates that the first element in the aligned Q-value $\boldsymbol{Q}_\theta$ corresponds to an element in $\boldsymbol{Q}_\theta$, and the same is true for $j_i$. $T$ is the length of the alignment. $\boldsymbol{Q}_\theta'$ and $\hat{\boldsymbol{Q}}'$ are obtained by expanding and repeating elements according to $\pi$.

## A.6 LIMITATION AND FUTURE WORK

However, PSP has limitations, particularly in its reliance on a replay buffer for updating the sampling policy. If the data is highly sensitive, the security of the replay buffer becomes a critical issue, as any potential leakage could have serious consequences. On the other hand, our PSP has the potential for application in more complex downstream tasks. PSP can help save resources in tasks with high annotation costs, such as Human-Object Interaction detection and semantic segmentation, through further improvements. Therefore, this will be a focus of my future work.

## A.7 BROADER IMPACTS

**Positive societal impacts.** PSP adaptively learns a sampling policy in an End-to-End manner to select the most informative samples under a limited annotation budget, thereby reducing the reliance on large-scale labeled datasets for downstream tasks. PSP achieves comparable performance to prompt learning with a fully labeled dataset, without relying on domain-specific knowledge for sample selection. This highlights its strong potential for extension to other computer vision tasks, such as object detection and semantic segmentation, where it can substantially reduce the annotation burden in these traditionally resource-intensive tasks.

**Negative societal impacts.** Despite its benefits, the proposed PSP further alleviates the reliance on manual labeling by more efficiently identifying informative samples. However, this advancement may bring about unforeseen socioeconomic effects. Specifically, as reliance on human annotators decreases, especially for repetitive or low-complexity labeling tasks, there is a potential risk of reduced employment opportunities in the data annotation industry. This shift could disproportionately affect low-skilled workers whose livelihoods depend on such roles, potentially leading to job displacement and increased economic vulnerability in regions where annotation work is a key source of income.

