# OpenReview forum: "PSP: Prompt-Guided Self-Training Sampling Policy for Active Prompt Learning"
_ICLR.cc/2026/Conference — ICLR 2026 Poster_

### Official Review · Reviewer_dwtc · 2025-10-26

**Soundness:** 2
**Presentation:** 1
**Contribution:** 2
**Rating:** 6
**Confidence:** 3

**Summary:**

The paper proposes PSP, a Prompt-Guided Self-Training Sampling Policy for Active Prompt Learning with CLIP models. The method integrates a Vectorized Soft Actor-Critic Sampling Policy (VSSP) and an Uncertainty-Augmented Self-Training (UST) mechanism. VSSP introduces a real-pseudo hybrid reward and vectorized critics to guide sample selection using prompt information, while UST generates reliable pseudo-labeled data based on uncertainty and confidence. Extensive experiments on seven benchmark datasets show consistent gains over PCB and other active learning baselines.

**Strengths:**

1. The proposed PSP improves over strong PCB variants with clear gains in accuracy on multiple datasets, and the evaluation on on seven datasets is comprehensive.

2.  The paper provides detailed mathematical formulation for VSSP and UST mechanisms.

**Weaknesses:**

I am not an expert in active learning, so I might defer my final score after considering other reviewers’ comments. Nonetheless, I identify two main weaknesses:

1.	Writing structure and overly complex presentation.
The Introduction and Method sections could be reorganized to better highlight the motivation and intuition before introducing dense equations. Currently, the paper is overloaded with mathematical notations, which obscure the core ideas. The motivation for the proposed approach is not convincing, and the discussion on the limitations of prior work lacks depth.


2.	Practical clarity.
The method appears as an overly complicated mixture of existing techniques, which raises doubts about its real-world applicability and scalability.

**Questions:**

See Weaknesses.

**Details Of Ethics Concerns:**

N.A.

---

> ### Author Response · Authors · 2025-11-20
> **Response to Reviewer dwtc**
>
> Thank you for your constructive comments and suggestions, and they are exceedingly helpful for us to improve our paper. In the following, we will respond to your comments point by point.
>
> >**W1:** Writing structure and overly complex presentation. The Introduction and Method sections could be reorganized to better highlight the motivation and intuition before introducing dense equations. Currently, the paper is overloaded with mathematical notations, which obscure the core ideas. The motivation for the proposed approach is not convincing, and the discussion on the limitations of prior work lacks depth.
>
> **A:**
> We sincerely apologize for the inconvenience caused by our writing. In response, we have reorganized the Introduction and Method sections in the revised manuscript to better highlight the motivation and intuition before delving into the dense equations, and deepened the discussion on the limitations of prior work in the updated Introduction and Related Work sections.
>
> Firstly, we reclaim that our motivation lies in explicitly introducing prompts to guide sample selection toward those that facilitate the optimization of the prompt template, and fully exploiting the complementary information in the unselected samples. Moreover, we have simplified the mathematical notations, like updated Equation 2.
>
> Secondly, prior works treat sample selection and prompt learning as two decoupled, discrete stages, with a lack of explicit connection between them, which makes the active prompt learning task fragmented. In contrast, our method bridges these two stages by refining the sample policy through a customized reward derived from the prompt learning process, thereby explicitly leveraging the prompt to guide sample selection.
>
> >**W2:** Practical clarity. The method appears as an overly complicated mixture of existing techniques, which raises doubts about its real-world applicability and scalability.
>
> **A:** We appreciate your attention to practical clarity, particularly regarding the real-world applicability and scalability of our proposed method.
>
> Firstly, we would like to clarify that the proposed method is not an overly complicated mixture of existing techniques. Given that active prompt learning inherently integrates two existing techniques (i.e., sample selection and prompt learning), we deliberately design a clear and well-structured modular framework consisting of VSSP and UST. VSSP formulates sample selection as a reinforcement learning process and refines its sampling policy to identify the most informative samples for prompt learning, while UST generates reliable pseudo-labeled data based on the uncertainty and confidence to enrich model’s understanding of the overall data distribution.
>
> Secondly, we have demonstrated the real-world applicability and scalability of our well-structured method by conducting experiments on multiple real-world datasets in the Table 1 of revised manuscript. The simplified results are shown below, where real-world datasets such as Oxford Pets, Stanford Cars, Caltech101, and Aircraft, etc., encompass diverse and representative visual categories.
>
> | Method|DTD|Oxford Pets|EuroSAT|Flowers102|Caltech101|Stanford Cars|Aircraft|Average Acc|
> |-------------------|--------------|------------|-------------|-------------|-------------|---------------|-------------|----------------|
> |PCB(Entropy)|59.50 | 80.94 | 80.75 | 96.94 | 93.48 | 68.93 | 27.58 | 72.59|
> |PCB(Coreset)|56.38 | 79.50| 79.28 | 92.33 | 91.70 | 65.75 | 26.22 | 70.17|
> |PCB(BADGE) |62.33|83.16 | 81.50 |96.71| 93.85 | 70.70 | 32.27 |74.36|
> | PSP | 65.66| 86.57 | 85.43 | 96.35 | 93.87| 73.84| 36.42 | 76.87|
>
> We can notice that our method with all components operate synergistically surpasses baselines at least **2.51**% in terms of average accuracy across seven dataset and almost outperforms other methods in all real-word datasets. These results strongly prove the real-world applicability and scalability of our method.
>
> Moreover, we further substantiate the scalability of our method in Table 7 of the revised manuscript by applying PSP to other pretrained VLMs, such as SigLIP, to demonstrate its scalability. The experimental results shown below indicate that PSP consistently outperforms PCB across four real-world datasets. These findings confirm that our method exhibits strong scalability across different pretrained VLMs.
>
> | Method| DTD| Oxford Pets | EuroSAT | Aircraft | Average Acc |
> |---------------|-------|------------|---------|---------|-------------|
> | PCB (SigLIP)| 55.73 | 47.12| 69.46| 22.98| 48.82|
> | PSP (SigLIP)| 59.75 | 62.99| 86.18| 28.65| 59.39|

---

> ### Comment · Reviewer_dwtc · 2025-11-26
>
> Thanks for the authors’ responses. I am glad to see that additional motivations were provided in the revision. However, I must note that I do not have a strong background in active learning, which makes it difficult for me to fully assess the method and its novelty. Therefore, I will keep my original overall score as weak accept but increase the presentation score.
>
> A personal suggestion is that, if possible, adding a short discussion or preliminary section that introduces existing active learning directions and compares their strengths and weaknesses could help reviewers like me—who lack background in this area—better follow the paper. That said, this is only a suggestion; experts in this specific research direction may not need such additional context.

---

> > ### Author Response · Authors · 2025-11-27
> > **Response to Reviewer dwtc regarding the preliminary section**
> >
> > Thank you for your insightful comment regarding a preliminary section that introduces existing active learning directions and compares their strengths and weaknesses. In response, we have added the preliminary section to Appendix A4 in new revision due to space limitations in the main text, and highlighted the new content in blue. The revisions are presented in the following.
> >
> > ***Active Learning** identifies criteria for selecting the most informative samples under a limited labeling budget. Active learning methods are primarily applied in three distinct scenarios: membership query synthesis (Mahapatra et al., 2018; Mayer & Timofte, 2020), stream-based (Narr et al., 2016; Fang et al., 2017; Woodward & Finn, 2017), and recently most mainstream pool-based (Vijayanarasimhan& Grauman, 2011; Gosselin & Cord, 2008; Yang et al., 2015; Kapoor et al., 2010; Bang et al., 2024) setting. In this work, we follow the pool-based setting and define an unlabeled data pool from which a subset of samples is actively selected for annotation. Based on the criteria, active learning methods can be categorized into three main approaches: Uncertainty-based sampling (Gal et al., 2017; Wang et al., 2019), Diversity-based sampling (Hacohen et al., 2022; Shui et al., 2020), Hybrid sampling (Ash et al., 2020; Parvaneh et al., 2022; Caramalau et al., 2021), and RL-based sampling (Ash et al., 2020; Kirsch et al., 2019). **Uncertainty-based sampling**, a straightforward and effective strategy, focuses on selecting samples that the model struggles to learn, employing techniques such as Monte-Carlo Dropout (Gal et al., 2017; Kirsch et al., 2019), Entropy (Holub et al., 2008), and Least Confident (Lewis & Catlett, 1994). Holub et al. proposed Entropy (Holub et al., 2008) for object recognition, which selects the samples with the highest entropy for annotation. **Diversity-based sampling** focuses on selecting samples that represent the full data distribution to ensure diversity in the labeled data, including clustering (Hu et al., 2021) and Coreset (Sener & Savarese, 2018). Sener et al. introduced the elegant and mathematically rigorous Coreset (Sener & Savarese, 2018), which provides an approximate upper bound on the loss for feature space coverage-based active learning algorithms. **Hybrid sampling** (Ash et al., 2020; Parvaneh et al., 2022) takes into account both diversity and uncertainty, aiming to mitigate the issue of redundancy in Uncertainty-based sampling and the limitations of Diversity-based sampling, where basic feature coverage strategies may fall short in assessing the model’ s confidence in its predictions. ALFA-Mix (Parvaneh et al., 2022) ultilizes unlabeled data to support active learning by interpolating between the representations of labeled and unlabeled instances and identifying features the model fails to recognize through inconsistencies in predicted labels. However, Hybrid sampling relies on fixed rules to balance diversity and uncertainty, limiting its adaptability across tasks. **RL-based sampling** formulates a sample selection policy, where Reinforcement Learning (RL) is applied to learn a policy that maximizes cumulative reward by selecting samples. Woodward et al. developed AOL (Woodward & Finn, 2017) that combines meta-learning and reinforcement learning for one-shot classification tasks. Liu et al. Introduced DRAL (Liu et al., 2019) to guide an agent in acquiring pairwise annotated data. Notably, PAL (Fang et al., 2017) builds a deep Q-network as an adaptive policy for sample selection. Therefore, we believe that RL-based methods have the potential to incorporate prompts for guiding sample selection. However, AOL (Woodward & Finn, 2017) and PAL (Fang et al., 2017) model the decision of whether to annotate a streaming unlabeled sample as a binary classification problem, while MedSelect (Vrabac et al., 2022) and DARL (Liu et al., 2019) rely on pairwise data, making them unsuitable for direct application in Active Prompt Learning (APL). Therefore, we introduce Soft Actor-Critic (SAC) (Haarnoja et al., 2018), a representative reinforcement learning algorithm known for its robustness to hyperparameters and strong performance in continuous action spaces. By designing a customized real-pseudo hybrid reward and vectorized critics, SAC can be seamlessly integrated into APL*

---

### Official Review · Reviewer_aJjs · 2025-10-26

**Soundness:** 3
**Presentation:** 2
**Contribution:** 2
**Rating:** 4
**Confidence:** 4

**Summary:**

This paper proposes PSP, an active prompt learning framework for CLIP to reduce annotation costs. PSP uses the prompt to guide sample selection and considers selected and unselected data. Moreover, PSP introduces two components: VSSP, an RL policy with a prompt-derived reward function to select informative samples, and UST, a self-training mechanism that generates and filters reliable pseudo-labels from unselected data.

**Strengths:**

1. The conceptual integration of prompt learning with RL is a strength.

2. The paper's claims are substantiated by extensive experiments across various aspects.

**Weaknesses:**

1.The paper suffers from several clarity issues that hinder understanding. Several symbols are used before definition. For example, $K$ (presumably the number of classes) is used on 213 without definition. $\hat{y}_i^k$ (Eq. 5) is not explicitly defined, though it seems to be the teacher CLIP's prediction if $k=u$. This pattern of undefined symbols increases reading difficulty.

2.The paper defaults to SAC (a 2018 algorithm) without justifying this choice over simpler selection methods (e.g., a learned Mixture-of-Experts) or newer, more advanced RL algorithms. The specific advantages of SAC for this particular selection problem are not discussed.

3.The equation (2) seems redundant with the text on 215(e.g.,  $f_{V}^{t,i}=F_{V}^{t}(x_{i}^{u})$  ).

4.The principle behind (2),(4),(5) is not explained.

5.Table 8: Comparing PSP to CoOp and CoCoOp is problematic. (a) These are outdated baselines; newer methods like [1]MaPLe or [2]PromptSRC can be used. (b) The comparison is fundamentally unfair, as PSP uses both labeled and unlabeled data, while CoOp/CoCoOp only "using the same number of real-labeled training samples". A fair comparison would require augmenting CoOp/CoCoOp with a pseudo-labeling strategy or a direct comparison against unsupervised methods, like [3]PromptKD.

6.Comparing a task-adapted PSP (fine-tuned on downstream data) to zero-shot VLMs is an apples-to-oranges comparison. This does not prove PSP is better. A valid comparison would be to apply the PSP to these other VLMs to demonstrate its adaptability, as was done for SigLIP (Table 7).

7.The dataset (presumably DTD) in Table 9 is not explicitly named. The numbers under the rounds (e.g., "305, 380...") are undefined. What do they represent? I am curious about the additional training time (or computational overhead) incurred by PSP compared to methods like CoOp.

8.The paper frequently refers to "student CLIP" and "teacher CLIP". The CLIP model's parameters are frozen; only the prompts are being trained. The correct terms may be "student prompt" and "teacher prompt" to reflect what is actually being updated.

[1]MaPLe: Multi-modal Prompt Learning

[2]Self-regulating Prompts: Foundational Model Adaptation without Forgetting

[3]PromptKD: Unsupervised Prompt Distillation for Vision-Language Models

**Questions:**

See weaknesses.

---

> ### Author Response · Authors · 2025-11-20
> **Response to Reviewer aJjs (1/2)**
>
> Thanks for your encouraging words and constructive comments. We sincerely appreciate your time in reading the paper, and our point-to-point responses to your comments are given below.
>
> >**W1:** The paper suffers from several clarity issues that hinder understanding. Several symbols are used before definition. For example, $K$ (presumably the number of classes) is used on 213 without definition. $\hat{y}_i^k$(Eq. 5) is not explicitly defined, though it seems to be the teacher CLIP's prediction if $k=u$. This pattern of undefined symbols increases reading difficulty.
>
> **A:** We sincerely apologize for any confusion caused by our writing. In response, we have added the definition of $K$ in 221 line of revised manuscript. Meanwhile, we fully acknowledge that $\hat{y}_i^u$ is the teacher CLIP' s prediction and have provided the explicit definition regarding $\hat{y}_i^u$ near Eq. 5 of revised paper.
>
> >**W2:** The paper defaults to SAC (a 2018 algorithm) without justifying this choice over simpler selection methods (e.g., a learned Mixture-of-Experts) or newer, more advanced RL algorithms. The specific advantages of SAC for this particular selection problem are not discussed.
>
> **A:**
> We greatly appreciate your constructive questions. In response, we will introduce the advantages of SAC in selection problem over newer RL algorithms and a learned Mixture-of-Experts (MoE), respectively.
>
> First, SAC is naturally better suited to Active Prompt Learning (APL) with limited experiences compared with newer RL algorithms, as it has proven high sample efficiency and strong performance across various domains [4, 12, 13], making it a reliable choice for scenarios with limited experiences. Specifically, SAC maintains an additional experience replay buffer compared with newer RL algorithms (e.g., OPO [9], RPO[10]), which enables the sampling policy to effectively reuse past experiences, achieving higher sample efficiency with limited experiences. Meanwhile, SAC retains the entropy regularization term, unlike EVT [11], which prevents the policy from overfitting to the limited experiences and ensures stable policy updates. Moreover, we compare with newer RL algorithm RPO and OPO on DTD. We notice that SAC outperforms RPO and OPO up to **2.54**% and **3.07**%, respectively. The results prove that SAC is more compile with APL.
>
> Second, a learned MoE is fundamentally unsuitable for APL, because MoE-based sample selection and prompt learning occur as two discrete, decoupled stages, preventing the MoE from updating its parameters in accordance with prompt learning’ s objective. In contrast, SAC has the potential to refine the sampling policy through a customized reward derived from the prompt learning process.
>
> >**W3:** The equation (2) seems redundant with the text on 215 (e.g., $f_V^{t, i} = F_V^t(x_i^u)$).
>
> **A:**
> Thank you for your careful reading and for pointing out the redundant expression. In response, we have replaced $F_V^t(x_i^u)$ with $f_V^{t,i}$ in updated Eq. 2 to simplify the formula.
>
> >**W4:** The principle behind (2),(4),(5) is not explained.
>
> **A:**
> Thank your for your valuable feedback. We have explained the principle behind these equation in revised paper.
>
> For Eq. 2, our principle is incorporating the classification score information of unlabeled samples to enrich the state representation and enable more effective learning of the sampling policy. To verify its effectiveness, we have conducted experiments using only features of unlabeled samples as state in updated Section 4.3. The results indicate that using only features as state in PSP leads to a performance decrease from **65.66**% to **63.36**% compared with using gradient embeddings. We conclude that modeling the state with classification score information helps refine the sampling policy.
>
> For Eq. 4, our principle is that the reward introduces MS indicator as a weight factor for performance-based reward, inspired by works [14, 15, 16], to enhance the link with the construction of real-labeled data. To validate its effectiveness, we remove the MS indicator and obtain the following results. We can notice that removing the MS indicator results in a performance drop of **1.83%**, proving that the reward with the MS indicator facilitates the update of the sampling policy. Additionally, the reward in Eq. 4 provides more comprehensive feedback for the sampling policy by incorporating pseudo-labeled rewards. When the pseudo-labeled reward is removed, the performance decreases by **1.77%**, highlighting the effectiveness of pseudo-labeled rewards.
>
> | Method| Final Acc |
> | :---------------- | --- |
> | PSP (w/o MS indicator)|63.83|
> | PSP (w/o pseudo-labeled reward)| 63.89|
> | PSP| 65.66 |
>
> For Eq. 5, we adhere to the principle that the reward should reflect the differences in performance for each individual sample. Specifically, a larger reward for a single sample in Eq. 5 indicates a worse prediction for that sample.

---

> ### Author Response · Authors · 2025-11-20
> **Response to Reviewer aJjs (2/2)**
>
> >**W5:** Table 8: Comparing PSP to CoOp and CoCoOp is problematic. (a) These are outdated baselines; newer methods like [1]MaPLe or [2]PromptSRC can be used. (b) A fair comparison would require augmenting CoOp/CoCoOp with a pseudo-labeling strategy or a direct comparison against unsupervised methods, like [3]PromptKD.
>
> **A:**
> We sincerely apologize for the problematic comparison with CoOp and CoCoOp. In response, we conduct two additional experiments following your rigorous guidance.
>
> - (a) We have compared PSP with MaPLe and PromptSRC, and the results are shown below. We observe that PSP achieves higher performance than these newer methods, proving that PSP is an effective method by actively annotating samples and fully utilizing the complementary information within unlabeled data.
>
> | Method| DTD|EuroSAT|
> | :--------| --- | --- |
> |MaPLe| 51.30 | 60.88 |
> |PromptSRC| 65.31|84.73 |
> | PSP| 65.66 | 85.43 |
>
> - (b) We conduct experiments comparing PSP with PromptKD and augmented CoOp. In these experiments, we enhance CoOp with our UST module and include additional unsupervised methods such as UPL [8] for a more comprehensive comparison. The results shown below indicate that PSP consistently outperforms other methods, demonstrating that PSP effectively facilitates prompt optimization.
>
> | Method| DTD| EuroSAT|
> | :--------| --- | --- |
> |Augmented CoOp| 61.11 | 79.94 |
> |PromptKD|60.93|49.21|
> |UPL|46.10|52.17|
> |UPL*|55.08|71.04|
> |PSP|65.66 |85.43|
>
> >**W6:** Comparing PSP with zero-shot VLMs is an apples-to-oranges comparison. This does not prove PSP is better. A valid comparison would be to apply the PSP to these other VLMs to demonstrate its adaptability, as was done for SigLIP (Table 7).
>
> **A:**
> We sincerely apologize for the abundant comparison between PSP with zero-shot VLMs. We would like to remove these experiments in our updated manuscript. Thank your for pointing out a valid comparison in Table 7 to demonstrate PSP’ s adaptability.
>
> >**W7:** The dataset (presumably DTD) in Table 9 is not explicitly named. The numbers under the rounds are undefined. I am curious about the additional training time incurred by PSP compared to methods like CoOp.
>
> **A:**
> Thank you for raising these important questions. First, we have explicitly specified the dataset in Table 9 as DTD and clarified that the numbers under each round (e.g., "305, 380...") represent the per-round training time (in seconds) in revised Appendix A.3. Second, the additional training time incurred by PSP compared to CoOp is **0.781**h, where the total training time of PSP and CoOp are **1.908**h and **1.127**h, respectively. The extra time of PSP over CoOp is spent on actively selecting the most informative samples for annotation in each round, achieving a performance gain from **58.77**% to **65.66**%.
>
> >**W8:** The paper frequently refers to "student CLIP" and "teacher CLIP". The CLIP model's parameters are frozen; only the prompts are being trained. The correct terms may be "student prompt" and "teacher prompt" to reflect what is actually being updated.
>
> **A:**
> Thank you for your thoughtful suggestion. We would like to clarify the reason why we adopt the terms “student CLIP” and “teacher CLIP”.
>
> First, our method is built upon a teacher–student paradigm, in which the complete CLIP model is regarded as an integrated information-processing unit. Replacing the original terms entirely with “student prompt” and “teacher prompt” would conceptually separate the prompts from the CLIP backbone, making it difficult to clearly describe the two functional branches—one responsible for updating the sampling policy and the other for generating pseudo labels.
>
> Second, our terminology follows the conventions adopted in related works such as [5, 6, 7], which also train only the prompts while keeping two encoders frozen.
>
> [1]MaPLe: Multi-modal Prompt Learning
>
> [2]Self-regulating Prompts: Foundational Model Adaptation without Forgetting
>
> [3]PromptKD: Unsupervised Prompt Distillation for Vision-Language Models
>
> [4]SAC-MS: Joint Slice Resource Allocation, User Association and UAV Trajectory Optimization with No-Fly Zone Constraints
>
> [5]Cascade Prompt Learning for Vision-Language Model Adaptation
>
> [6]Hierarchical Knowledge Prompt Tuning for Multi-task Test-Time Adaptation
>
> [7]Improving Zero-shot Generalization of Learned Prompts via Unsupervised Knowledge Distillation
>
> [8]Unsupervised prompt learning for vision-language models
>
> [9]On-Policy RL with Optimal Reward Baseline
>
> [10]Reflective Policy Optimization
>
> [11]Extreme Q-Learning: MaxEnt RL without Entropy
>
> [12]Sac-based computation offloading and resource allocation in vehicular edge computing
>
> [13]Few-Shot Preference Learning for Human-in-the-Loop RL
>
> [14]Learning Rewards to Optimize Global Performance Metrics in Deep RL
>
> [15]Multi-Agent Reinforcement Learning with Common Policy for Antenna Tilt Optimization
>
> [16]Optimizing Language Models with Fair and Stable Reward Composition in Reinforcement Learning

---

> > ### Comment · Reviewer_aJjs · 2025-11-22
> >
> > Thank you for the comprehensive response. I appreciate the extensive experiments and clarifications provided. However, I still have some concerns:
> >
> > - **Inefficiency compared to PromptSRC:** Does PromptSRC in your experiment use the exact same number of real-labeled samples as your approach?  As you stated in your paper "CoOp and CoCoOp using the same number of real-labeled training samples as our approach."  If so, the results (PSP $\approx$ PromptSRC) imply that your method is inefficient.  PSP utilizes both **labeled and unlabeled data** (plus the overhead of active selection), yet it barely matches PromptSRC, which uses less data.
> >
> > - **Questionable PromptKD Implementation:** PromptKD, as an unsupervised method, should theoretically utilize a data scale comparable to your total pool (labeled + unlabeled).  In existing literature, PromptKD typically outperforms PromptSRC significantly.  However, your reported result for PromptKD is anomalously low—even worse than CoOp.  This strongly suggests an incorrect implementation or unfair setting for PromptKD, which invalidates the comparison.

---

> > > ### Author Response · Authors · 2025-11-24
> > > **Response to Reviewer aJjs' s two concerns**
> > >
> > > Thank you for your encouraging words and constructive feedback. We sincerely appreciate the time you’ve spent reviewing our paper, and below are our point-by-point responses to your comments.
> > >
> > > >**Q1:** Inefficiency compared to PromptSRC: Does PromptSRC in your experiment use the exact same number of real-labeled samples as your approach? As you stated in your paper "CoOp and CoCoOp using the same number of real-labeled training samples as our approach." If so, the results (PSP≈PromptSRC) imply that your method is inefficient. PSP utilizes both labeled and unlabeled data (plus the overhead of active selection), yet it barely matches PromptSRC, which uses less data.
> > >
> > >
> > >
> > > **A:**
> > > Sincerely thank you for your detailed reading and thoughtful consideration. As noted in your comment, PromptSRC indeed uses the same number of real-labeled samples as the proposed method in our experiment. However, we would like to clarify that the results (PSP ≈ PromptSRC) should not imply that the method is inefficient.
> > >
> > > Specifically, we reclaim that our method is built upon the emerging paradigm of active prompt learning, which actively annotates unlabeled samples to maximize the performance of prompt learning within a limited budget — significantly differing from traditional prompt learning methods. To demonstrate the effectiveness of our method, we do not use any unlabeled data during prompt learning and replace the default CoOp for prompt learning with PromptSRC, denoted as VSSP (PromptSRC). The comparison results are shown below. We observe that our VSSP significantly improves the performance of PromptSRC, achieving **3.86**% improvement in EuroSAT and **2.60**% improvement in DTD. These results strongly demonstrate that our VSSP effectively enhances the performance of prompt learning. Given that all components in VSSP are lightweight, consisting of only three layers of fully connected neural networks, the performance gain further emphasizes that our method is efficient in actively annotating the most informative samples for existing prompt learning methods like PromptSRC.
> > >
> > > | Method| DTD|EuroSAT|
> > > | :--------| --- | --- |
> > > |PromptSRC| 65.31|84.73 |
> > > |VSSP (PromptSRC)|67.91|88.59|
> > >
> > > We hope these additional experiments and clarifications address your concern and highlight the effectiveness of our method. Thank you again for your constructive feedback.
> > >
> > > >**Q2:** Questionable PromptKD Implementation: PromptKD, as an unsupervised method, should theoretically utilize a data scale comparable to your total pool (labeled + unlabeled). In existing literature, PromptKD typically outperforms PromptSRC significantly. However, your reported result for PromptKD is anomalously low—even worse than CoOp. This strongly suggests an incorrect implementation or unfair setting for PromptKD, which invalidates the comparison.
> > >
> > > **A:**
> > > Thank you for your insightful and valuable comments. We fully acknowledge that PromptKD, as an unsupervised method, utilizes a data scale comparable to our total pool.
> > >
> > > In response, we would like to clarify that the reported anomalously low result for PromptKD (even performing worse than CoOp) is directly attributed to the fundamental difference in the experimental setups: PromptKD is a purely unsupervised prompt learning method, whereas CoOp and PromptSRC in our context utilize labeled data for supervised training. This difference fundamentally changes the basis of comparison.
> > >
> > > First, the unsupervised prompt learning in PromptKD is achieved by first pre-training a teacher model, and then performing logits-level knowledge distillation with the teacher model. We follow the setup from Table 2 in the original PromptKD paper, where the teacher model is pre-trained on ImageNet, and unsupervised prompt learning is performed using unlabeled training samples from DTD and EuroSAT. Our results for PromptKD significantly outperform the reported performance of PromptSRC in the original PromptKD paper by **14.06**% in DTD and **3.71**% in EuroSAT in this original source (ImageNet) to target (DTD or EuroSAT) setting, which aligns well with your comment that PromptKD typically outperforms PromptSRC significantly.
> > >
> > > However, in this original source-to-target setting, PromptSRC in the original PromptKD paper did not utilize any training samples from DTD and EuroSAT. In contrast, PromptSRC and CoOp reported in our response are trained in a supervised manner using the labeled training data available in DTD and EuroSAT. Therefore, the performance of unsupervised PromptKD is naturally lower than that of supervised PromptSRC and CoOp in our experiments.
> > >
> > > We hope this detailed explanation resolves your concerns. Thank you once again for your valuable feedback, which has guided us to a deeper understanding of the experimental phenomena.

---

### Official Review · Reviewer_gdcF · 2025-10-28

**Soundness:** 3
**Presentation:** 3
**Contribution:** 4
**Rating:** 8
**Confidence:** 3

**Summary:**

This paper proposes a new method called PSP for the field of Active Prompt Learning (APL). Its main goal is to efficiently train prompts for vision-language models (such as CLIP) under a limited annotation budget by intelligently selecting the most valuable data samples for labeling, thereby adapting the prompts to downstream tasks.

**Strengths:**

[1] The experiments in the paper are comprehensive.
[2] The challenging decision of “which samples to select for labeling” is modeled as a reinforcement learning task. This allows the sampling policy (Actor) to act like an agent that continuously “learns” and “evolves” through interaction with the environment (model performance), ultimately mastering the most efficient sampling strategy. This approach is more flexible and intelligent than traditional methods that rely on fixed rules, such as uncertainty or diversity.
[3] The paper combines active learning techniques for vision-language models with reinforcement learning, effectively improving the efficiency of sample learning.

**Weaknesses:**

[1] The algorithm incurs excessive computational overhead.
[2] The system is relatively complex, and parameter tuning may be challenging.

**Questions:**

[1]  In such “low-data” or “few-episode” reinforcement learning scenarios, how does your method ensure that the Actor policy effectively converges rather than overfitting to the limited early experiences? Does your design of the Vectorized Critic play a key role here, as it can provide richer learning signals for multiple samples from a single experience?
[2] Have you analyzed how the learned policy behaves compared to traditional active learning heuristics, such as uncertainty sampling, diversity sampling, or hybrid methods like BADGE? For instance, does the model prefer high-uncertainty samples at certain stages while prioritizing diversity at others? Can we reverse-engineer the learned “optimal policy” to extract interpretable new sampling insights that go beyond existing heuristic rules?

---

> ### Author Response · Authors · 2025-11-20
> **Response to Reviewer gdcF (1/2)**
>
> Thanks for your encouraging words and constructive comments. We sincerely appreciate your time in reading the paper, and our point-to-point responses to your comments are given below.
>
> >**W1:** The algorithm incurs excessive computational overhead.
>
> **A:** We sincerely appreciate your insightful feedback. We would like to clarify that our PSP slightly reduces computational overhead compared with PCB, while still maintaining performance improvements. To support this, we report the total training time, total parameters and final accuracy on DTD in the following table.
>
> | Method| Training Time (h) |Params (M)| Final Acc (%) |
> |--------------|-------------------|-------|----------------|
> | PCB| 1.95|127.56| 62.33|
> | PSP| 1.91|128.35|65.66|
>
> We can observe that PSP outperforms PCB by up to **3.33**% in final accuracy while also saving **0.04**h of training time and maintaining similar total parameter count. The results reveal that PSP strikes a balance between model efficiency and performance improvement.
>
> >**W2:** The system is relatively complex, and parameter tuning may be challenging.
>
> **A:** Thank you for pointing out the potential concern regarding system complexity and parameter tuning.
>
> First, as noted in your comment, our system may appear relatively complex at first glance, but it follows a clear and well-structured modular design to support the Active Prompt Learning (APL) task. Given that APL inherently integrates two existing components (i.e., sample selection and prompt learning), we deliberately design a well-structured framework consisting of VSSP and UST. VSSP formulates sample selection as a reinforcement learning process and refines its sampling policy to identify the most informative samples for prompt learning, while UST generates reliable pseudo-labeled data based on the uncertainty and confidence to enrich model’s understanding of the overall data distribution.
>
> Second, we would like to clarify that parameter tuning of our system is not challenging, as VSSP is tightly integrated with the system and the backbone of the student CLIP is frozen. Specifically, the parameter tuning of VSSP is driven by the reward signal from the student CLIP’s prompt learning, which guides the updates of the lightweight networks within VSSP. Moreover, the student CLIP is equipped only with a small set of parameters within a learnable prompt, while UST does not include any trainable parameters.
>
> >**Q1:** In such “low-data” or “few-episode” reinforcement learning scenarios, how does your method ensure that the Actor policy effectively converges rather than overfitting to the limited early experiences? Does your design of the Vectorized Critic play a key role here, as it can provide richer learning signals for multiple samples from a single experience?
>
> **A:**
> We greatly appreciate your thorough feedback. In response, we will address the raised questions one by one.
>
> First, we ensure that the Actor policy achieves effective convergence and avoids overfitting to the limited early experiences through replay buffer and entropy regularization in Equation 7, following prior works [1, 2, 3]. Specifically, replay buffer continually accumulates experiences across rounds and allows the Actor policy to reuse these experiences generated by the old policy, thus effectively preventing the policy from overfitting to the limited early experiences. Moreover, entropy regularization encourages the sampling policy to maintain a certain level of stochasticity and explore a wider range of actions, thereby preventing overfitting.
>
> Second, as mentioned in your comment, the Vectorized Critic indeed plays a key role in our method by providing richer learning signals for multiple samples from a single experience. It can generate multiple value estimates from a single experience, thereby providing the Actor policy with richer and more fine-grained learning signals. This design substantially enhances sample efficiency, offers smoother and more stable gradient feedback, and effectively mitigates the risk of overfitting to limited early experiences. Moreover, we replace the Vectorized Critic with the standard Critic, which leads to a **1.24**% performance drop on DTD, further proving its effectiveness.

---

> ### Author Response · Authors · 2025-11-20
> **Response to Reviewer gdcF (2/2)**
>
> >**Q2:** Have you analyzed how the learned policy behaves compared to traditional active learning heuristics, such as uncertainty sampling, diversity sampling, or hybrid methods like BADGE? For instance, does the model prefer high-uncertainty samples at certain stages while prioritizing diversity at others? Can we reverse-engineer the learned “optimal policy” to extract interpretable new sampling insights that go beyond existing heuristic rules?
>
> **A:**
> Thank you for raising these important questions. In response, we aim to answer these questions sequentially.
>
> First, we have analyzed how the learned policy behaves compared to traditional active learning heuristics on DTD in updated Appendix A3. Specifically, we compare our sampling policy with BADGE, Entropy, and Coreset, and report the overlap ratio, computed as the intersection between the samples selected by both methods divided by the number of samples selected by our sampling policy.
>
> | Round | PSP vs Coreset | PSP vs Entropy | PSP vs BADGE |
> |-------|----------------|----------------|----------------|
> | 1 | 0.0113 | 0.0113 | 0.0113 |
> | 2 | 0.1107 | 0.0959 | 0.0923 |
> | 3 | 0.0920 | 0.0920 | 0.0958 |
> | 4 | 0.1103 | 0.0608 | 0.0837 |
> | 5 | 0.1094 | 0.1283 | 0.1283 |
> | 6 | 0.1301 | 0.0892 | 0.0706 |
> | 7 | 0.0784 | 0.1007 | 0.1045 |
> | 8 | 0.0833 | 0.1023 | 0.1023 |
>
> The results above show that PSP exhibits relatively high consistency with Coreset in the early rounds (Rounds 2-4), while in later rounds (Rounds 5-8), its overlap with Entropy increases. This indicates that the sampling policy initially favors diverse samples to establish broad coverage, and gradually shifts to uncertain samples as the model’ s classification capability improves. Notably, since the overlap between PSP and BADGE is not consistently higher than that with the other heuristics, we can conclude that PSP is not merely imitating a fixed hybrid rule such as BADGE. Instead, it continuously interacts with the prompt learning process and adaptively adjusts its sampling policy over rounds.
>
> Second, we further reverse-engineer the learned “optimal policy” from the above results to derive interpretable sampling insights, suggesting that sample selection should prioritize diversity in the early rounds to explore the input space and gradually shift toward uncertainty-based selection as the model’ s capability improves. However, the additional principles learned by the sampling policy beyond existing heuristic rules remain to be explored.
>
> We thank your again for inspiring this new research direction. In future work, we plan to further enhance the interpretability of the learned sampling policy.
>
> [1]Reinforcement Learning with Deep Energy-Based Policies
>
> [2]Soft Actor-Critic: Off-Policy Maximum Entropy Deep Reinforcement Learning with a Stochastic Actor
>
> [3]Prioritized experience replay

---

### Official Review · Reviewer_7H9h · 2025-11-01

**Soundness:** 3
**Presentation:** 3
**Contribution:** 3
**Rating:** 6
**Confidence:** 4

**Summary:**

This paper targets the problem of sample selection in Active Prompt Learning (APL) for Vision-Language Models (VLMs). The authors argue that existing APL methods use generic active learning heuristics that do not explicitly use information from the learned prompt to guide sample selection and also ignore valuable information in the large pool of unselected data. To address this, the paper proposes PSP (Prompt-Guided Self-Training Sampling Policy), a framework that learns a sampling policy using reinforcement learning (RL). The method consists of two main parts: (1) a Vectorized Soft Actor-Critic Sampling Policy (VSSP), where an RL agent learns to select informative samples based on a state representation of gradient embeddings and a custom "real-pseudo hybrid reward" that reflects the prompt's performance; and (2) an Uncertainty Augmented Self-Training (UST) mechanism that generates pseudo-labels for unselected data to augment the training set. Experiments on seven image classification datasets show that PSP outperforms several active learning baselines, including variants of the state-of-the-art PCB method.

**Strengths:**

1. Novel Formulation for Sample Selection: The core idea of framing active sample selection for prompt tuning as a policy learning problem is novel and conceptually appealing.
2. Explicit Integration of Prompt Information: A key strength is the explicit use of the learned prompt's performance to guide the sampling policy.
3. Leveraging the Full Unlabeled Dataset: The UST mechanism is a sensible addition that attempts to extract more value from the entire unlabeled dataset, rather than only focusing on the small subset of samples selected for annotation
4. Comprehensive Experimental Evaluation: The paper presents an extensive empirical study on seven datasets

**Weaknesses:**

1. Extreme System Complexity for Modest Performance Gains: The primary weakness of this work is the immense complexity of the proposed VSSP framework relative to the performance improvement it delivers.
2. Convoluted and Indirect Reward Function: The reward function in Equation 4, defined as $r(s_{t},a_{t})=log(p_{m}(g))*(\overline{r}_{s}+\beta\overline{r}_{p})$, is indirect and unintuitive.
3. Potential Instability and High Sample Complexity of RL: Actor-critic methods are known to be sensitive to hyperparameters and can be unstable to train, especially with limited data.

**Questions:**

1. The reward function is constructed as a product of the log-probability of the sampling scheme and the classification error. This objective is quite indirect. Have you experimented with a more direct reward signal, such as the negative validation accuracy on a small holdout set, or simply using $- (\overline{r}_{s} + \beta\overline{r}_{p})$ as the reward?
2. The VSSP agent is trained on a very small number of total experiences (a maximum of 8, one per round). Given the high sample complexity and potential instability of RL algorithms, how do you ensure that the learned policy is stable and generalizes beyond the few trajectories seen during training?
3. Could you provide a more detailed justification for using Soft-DTW to align the Q-value vectors between states $s_t$ and $s_{t+1}$?

---

> ### Author Response · Authors · 2025-11-20
> **Response to Reviewer 7H9h (1/2)**
>
> Thanks for your encouraging words and constructive comments. We sincerely appreciate your time in reading the paper, and our point-to-point responses to your comments are given below.
>
> >**W1:** Extreme System Complexity for Modest Performance Gains: The primary weakness of this work is the immense complexity of the proposed VSSP framework relative to the performance improvement it delivers.
>
> **A:**
> Thank you for your valuable feedback. In response, we would like to clarify that our VSSP is not overly complex and is effective in active prompt learning task. Specifically, we compare VSSP with the state-of-the-art method PCB on diverse datasets and report the total parameters, as summarized in the table below. We observe that VSSP achieves an average accuracy gain of **1.60**% across diverse datasets compared to PCB, while maintaining a similar total parameter count. We conclude that VSSP is not overly complex, with its all components consisting of three lightweight fully connected layers.
>
> | Method| DTD| Oxford Pets | EuroSAT | Aircraft | Average Acc |Params|
> |-------------------|-------|-------------|---------|----------|---------|------|
> | PCB| 62.33 | 83.16| 81.50| 32.27| 64.82|127.56 M|
> | VSSP| 63.77 | 85.55| 83.97| 32.40| 66.42|128.35 M|
>
>
> >**W2:** Convoluted and Indirect Reward Function: The reward function in Equation 4, defined as $r(s_t, a_t) = \log(p_m(g)) \cdot (\overline{r}_s + \beta \overline{r}_p)$, is indirect and unintuitive.
>
> **A:**
> Thank you for your insightful comments. We clarify that our reward introduces $\log(p_m(g))$ as a weight factor for performance-based reward, inspired by works [5, 6, 7], to strengthen the connection with the construction of real-labeled data, even though our reward itself is indirect and unintuitive. To elaborate this, we compare with other direct alternatives (i.e., $- (\overline{r}_{s} + \beta\overline{r}_{p})$) under your guidance in **Q1**. The results show that our reward outperform direct reward by **1.30**% in final accuracy, proving the effectiveness of our reward. The detailed analysis can be found in our response to **Q1**.
>
> >**W3:** Potential Instability and High Sample Complexity of RL: Actor-critic methods are known to be sensitive to hyperparameters and can be unstable to train, especially with limited data.
>
> **A:**
> Sincerely thank you for your valuable suggestion. We fully agree that RL, especially Actor-Critic methods are sensitive to hyperparameters and can be unstable to train with limited data. In response, we adopt maximize policy entropy, experience replay, and a target value network to initially alleviate potential instability and hyperparameter sensitivity, following prior works [1, 2, 3, 4].
>
> Specifically, we maximize the policy entropy to encourage exploration by making the sampling policy more stochastic, while simultaneously reducing VSSP’ s sensitivity to modeling and estimation errors, thereby improving overall robustness. Meanwhile, we introduce experience replay, which mitigates the bias caused by insufficient early samples and allows the Actor to obtain more stable updates by continually reusing experiences generated by old policies.
>
> Furthermore, we adopt the target value network to stabilize VSSP’ s training and prevent oscillation in critic learning, thus reducing sensitivity to critical hyperparameters such as target value smoothing coefficient $\tau$. Finally, we conduct a hyperparameter analysis on DTD regarding $\tau$ (default 0.01), as shown below. These results indicate that our method is not sensitive to variations in $\tau$.
>
> | $\tau$ | Final Acc |
> |---------|--------------|
> | 0.1| 65.39|
> | 0.01| 65.66|
> | 0.001|65.42|

---

> ### Author Response · Authors · 2025-11-20
> **Response to Reviewer 7H9h (2/2)**
>
> >**Q1:** The reward function is constructed as a product of the log-probability of the sampling scheme and the classification error. This objective is quite indirect. Have you experimented with a more direct reward signal, such as the negative validation accuracy on a small holdout set, or simply using $- (\overline{r} _ {s} + \beta\overline{r}_{p})$ as the reward?
>
> **A:**
> We greatly appreciate your valuable comment. In response, we clarify that our reward enhances the link to the construction of real-labeled data by incorporating the log-probability of the sampling scheme $\log(p_m(g))$ as weight of performance-based reward, despite it is indirect. To support our claim, we have compared our reward with direct reward $- (\overline{r}_{s} + \beta\overline{r} _ {p})$ and reported accuracy in each round.
>
> |Method|1|2|3|4|5|6|7|8|
> |---|---|---|---|---|---|---|---|---|
> |PSP (direct)|23.40|49.59|54.37|58.98|60.28|62.47|64.18|64.36|
> |PSP|23.82|48.46|58.81|58.92|61.52|62.12|63.95|65.66|
>
> We observe that replacing our reward with the direct reward results in a **1.30**% decrease in final accuracy. We conclude that our reward is more compatible with active prompt learning by incorporating $\log(p_m(g))$ as a weight factor, which is directly related to the construction of real-labeled data and reflects the quality of the sampling scheme. In future work, we will focus on designing a more effective and more direct reward under your guidance.
>
> >**Q2:** The VSSP agent is trained on a very small number of total experiences (a maximum of 8, one per round). Given the high sample complexity and potential instability of RL algorithms, how do you ensure that the learned policy is stable and generalizes beyond the few trajectories seen during training?
>
> **A:** Thank you for raising these critical points. As noted in your comments, VSSP is trained on a limited number of total experiences, determined by the default configuration in PCB. But we adopt entropy regularization, a replay buffer, and a target value network to ensure that the learned policy remains stable and generalizes beyond the few trajectories seen during training.
>
> Specifically, the entropy regularization in Equation 7 prevents the policy from overfitting to individual experiences. The replay buffer enables the policy to achieve more stable updates by repeatedly utilizing diverse experiences collected from old policies. The target value network enhances training stability in VSSP by providing smoothly updated and low-variance TD targets, thereby mitigating oscillatory behavior in critic learning.
>
> All these strategies collaboratively guarantee that the learned policy is stable and generalizes beyond limited experiences. To illustrate this, we analyze the final accuracy across seven datasets in updated Table 1. The results show that the proposed method consistently outperforms other models, proving the stability and generalizability of the learned sampling policy.
>
> >**Q3:** Could you provide a more detailed justification for using Soft-DTW to align the Q-value vectors between states $s_t$ and $s_{t+1}$?
>
> **A:**
> We are grateful for your insightful comment. We have provided a more detailed justification for using Soft-DTW near Equation 8 in the revised paper.
>
> First, the mechanism of Soft-DTW is well suited for the alignment of the Q-value vectors between states $s_t$ and $s_{t+1}$. (*i*) Soft-DTW measures the similarity between Q-value vectors from states $s_t$ and $s_{t+1}$ while preserving the relative order and structural relationships of their elements. This property is particularly important in our method, as the Q-value vectors from $s_t$ and $s_{t+1}$ are often strongly correlated. (*ii*) The differentiable nature of Soft-DTW allows gradients to propagate through the alignment operation, as shown in Equation 8. This contributes to more stable training and does not interfere with the update procedures of either the Actor or the Critic.
>
> Second, we have proven the effectiveness of Soft-DTW by comparing it with alternatives such as PAD and VAE, and reported the accuracy in each round as shown below. We observe that Soft-DTW improves the final accuracy by **1.83**% and **0.83**% compared to PAD and VAE, respectively.
>
> |Method|1|2|3|4|5|6|7|8|
> |---|---|---|---|---|---|---|---|---|
> |PSP (PAD)|37.94|52.24|58.69|60.16| 61.52|63.47|63.71|63.83|
> |PSP (VAE)|37.88|52.54|57.09|58.16|63.29|63.36|63.47|64.83|
> |PSP (Soft-DTW)|23.82|48.46|58.81|58.92|61.52|62.12|63.95|65.66|
>
> [1]Reinforcement Learning with Deep Energy-Based Policies
>
> [2]Soft Actor-Critic: Off-Policy Maximum Entropy Deep Reinforcement Learning with a Stochastic Actor
>
> [3]Prioritized experience replay
>
> [4]Deep Reinforcement Learning with Double Q-learning
>
> [5]Learning Rewards to Optimize Global Performance Metrics in Deep RL
>
> [6]Multi-Agent Reinforcement Learning with Common Policy for Antenna Tilt Optimization
>
> [7]Optimizing Language Models with Fair and Stable Reward Composition in
> Reinforcement Learning

---

### Author Response · Authors · 2025-11-20
**Changes present in revised version**

We would like to thank all reviewers for the insightful reviews and comments. Next, we record the following changes in the revised submission, with the changes highlighted in yellow:

- We have provided a more detailed justification for using Soft-DTW to align the Q-value vectors between states $s_t$ and $s_{t+1}$.

- We have analyzed how the learned policy behaves compared to traditional active learning heuristics.

- We have added the definition of $K$ at line 221 and provided an explicit definition of $\hat{y}_i^u$ near Equation 5.

- We have replaced $F_V^t(x_i^u)$ with $f_V^{t,i}$ in updated Equation 2 to simplify the formula.

- We have explained the principle behind Equation 2, 4 and 5.

- We have conducted experiments where only the features of unlabeled samples are used as the state in Section 4.3.

- We have explicitly specified the dataset in Table 9 as DTD and clarified that the numbers under each round (e.g., "305, 380...") represent the per-round training time (in seconds) in revised Appendix A.3.

- We have reorganized the Introduction and Method sections to better highlight the motivation.

- We have deepen the discussion on the limitations of prior work in the Introduction and Related Work sections.

---

### Meta-Review · Area_Chair_JyAJ · 2025-12-31

**Summary:**

This paper presents a Prompt-Guided Self-Training Sampling Policy (PSP) for APL. It integrates Soft Actor-Critic with a customized real-pseudo hybrid reward and vectorized critics to incorporate prompts in guiding sample selection toward those that facilitate the optimization of prompt template, by jointly considering both selected and unselected samples.

The reviewers generally appreciated the novelty of formulating sample selection as a policy learning problem and the comprehensive experimental evaluation across seven datasets. The authors provided a robust rebuttal, adding comparisons with newer methods (MaPLe, PromptSRC), clarifying the RL stability mechanisms, and refining the manuscript's structure. Considering the consistent performance improvements and the interesting intersection of RL and Prompt Learning, the decision is to accept.

**Reviewer Concerns:**

System Complexity and Computational Overhead (Reviewers 7H9h, gdcF, dwtc):

Reviewers were initially concerned that the RL-based approach added excessive complexity for modest gains.

Resolution: The authors clarified that VSSP uses lightweight networks and that the total training time and parameter count are comparable to (and in some cases more efficient than) the state-of-the-art baseline PCB. The trade-off between the slight architectural complexity and the performance gain (avg 1.6% over PCB) is considered acceptable.

Clarity and Notation (Reviewers aJjs, dwtc):

Concerns: Reviewers noted undefined symbols and dense mathematical notation.

Response: The authors revised the manuscript significantly, adding a preliminary section on Active Learning, defining all symbols (e.g., K, y_hat), and simplifying equations to improve readability.

**Reviewer Scores:**

Reviewer 7H9h: 6. This reviewer acknowledged the novelty and that the method works.

Reviewer gdcF: 8. This reviewer was highly positive about the "learning to sample" approach and the comprehensive experiments.

Reviewer aJjs: 4. This reviewer provided the most critical feedback regarding baselines. The authors' detailed responses regarding PromptKD and PromptSRC efficiency effectively defended the paper's technical merit.

Reviewer dwtc: 6. This reviewer admitted a lack of deep background in active learning but was convinced by the authors' improvements to the presentation and the clear empirical gains during the rebuttal.

---

### Decision · Program_Chairs · 2026-01-26

Accept (Poster)